# Structural dynamics of the active HER4 and HER2/HER4 complexes is finely tuned by different growth factors and glycosylation

Raphael Trenker[1†], Devan Diwanji[1,2†], Tanner Bingham[1,2], Kliment A Verba[3,4*], Natalia Jura[1,3,4*]

[1]Cardiovascular Research Institute, University of California, San Francisco, San Francisco, United States; [2]Medical Scientist Training Program, University of California, San Francisco, San Francisco, United States; [3]Department of Cellular and Molecular Pharmacology, University of California, San Francisco, San Francisco, United States; [4]Quantitative Biosciences Institute, University of California, San Francisco, San Francisco, United States

*For correspondence:
kliment.verba@ucsf.edu (KAV);
natalia.jura@ucsf.edu (NJ)

†These authors contributed equally to this work

**Abstract** Human Epidermal growth factor Receptor 4 (HER4 or ERBB4) carries out essential functions in the development and maintenance of the cardiovascular and nervous systems. HER4 activation is regulated by a diverse group of extracellular ligands including the neuregulin (NRG) family and betacellulin (BTC), which promote HER4 homodimerization or heterodimerization with other HER receptors. Important cardiovascular functions of HER4 are exerted via heterodimerization with its close homolog and orphan receptor, HER2. To date structural insights into ligand-mediated HER4 activation have been limited to crystallographic studies of HER4 ectodomain homodimers in complex with NRG1β. Here, we report cryo-EM structures of near full-length HER2/HER4 heterodimers and full-length HER4 homodimers bound to NRG1β and BTC. We show that the structures of the heterodimers bound to either ligand are nearly identical and that in both cases the HER2/HER4 heterodimer interface is less dynamic than those observed in structures of HER2/EGFR and HER2/HER3 heterodimers. In contrast, structures of full-length HER4 homodimers bound to NRG1β and BTC display more large-scale dynamics mirroring states previously reported for EGFR homodimers. Our structures also reveal the presence of multiple glycan modifications within HER4 ectodomains, modeled for the first time in HER receptors, that distinctively contribute to the stabilization of HER4 homodimer interfaces over those of HER2/HER4 heterodimers.

## eLife assessment

This manuscript describes structures of HER4 homo- and HER4/HER2 hetero-dimer complexes using single particle cryo-EM. This **important** work **convincingly** describes new structural details of these complexes that expand our understanding of their function. This work will be of interest to researchers working on cell surface signalling and kinase activity.

## Introduction

HER4 is a ubiquitously expressed receptor functioning in heart, mammary, and neural development (*Lemmon and Schlessinger, 2010*; *Yarden and Sliwkowski, 2001*; *Plowman et al., 1993*; *Muraoka-Cook et al., 2008*; *Gassmann et al., 1995*). Binding of extracellular growth factors leads to HER4

receptor homodimerization or heterodimerization with one of three other HER receptor family members, EGFR, HER2, or HER3, and subsequent activation of their intracellular kinase domains (*Lemmon and Schlessinger, 2010*; *Yarden and Sliwkowski, 2001*; *Plowman et al., 1993*). While HER4 activation is linked to signaling pathways activated by other HER receptors, including Ras/MAPK and PI3K/Akt, HER4 is the only HER with documented growth inhibitory effect on cells (*Muraoka-Cook et al., 2008*; *Sweeney and Carraway, 2000*; *Sweeney et al., 2000*). Consistent with this observation, and in contrast to other HER receptors for which genetic alterations are widely linked to oncogenesis (*Arteaga and Engelman, 2014*), HER4 is more commonly observed to be lost or downregulated in human cancers (*Muraoka-Cook et al., 2008*; *Arteaga and Engelman, 2014*; *Naresh et al., 2006*; *Segers et al., 2020*). More rarely, HER4-activating mutations and overexpression have been observed in lung, melanoma, and gastric cancers (*Segers et al., 2020*; *Prickett et al., 2009*).

HER4 plays distinct roles in the nervous and cardiovascular systems from other HER receptors, which are underscored by the pathological consequences of dysregulated HER4 signaling (*Gassmann et al., 1995*; *Mei and Nave, 2014*; *Bersell et al., 2009*). Aberrant activation of HER4 is associated with neurological diseases including amyotrophic lateral sclerosis (ALS), schizophrenia, and other psychological disorders, where inhibitory missense mutations in HER4, and either increased or decreased levels of the HER4 ligand NRG1, can lead to various disease phenotypes (*Mei and Nave, 2014*; *Takahashi et al., 2013*; *Song et al., 2012*). In cardiomyocytes, HER4 heterodimerization with HER2 is particularly important for survival under acute stress conditions (*Plowman et al., 1993*; *Bersell et al., 2009*; *Lee et al., 1995*). HER2 and HER4 signaling are both essential for embryonic and postnatal heart development (*Gassmann et al., 1995*; *Lee et al., 1995*; *Odiete et al., 2012*). As an orphan receptor, HER2 does not undergo ligand-induced homodimerization and relies on HER4 for activation (*Tzahar et al., 1996*; *Wallasch et al., 1995*) when HER4 is bound to NRG1 produced by the cardiac endothelium (*Odiete et al., 2012*; *Figure 1a*). Disruption of the HER2/HER4 signaling has been attributed to the cardiotoxic effects of HER2-targeting cancer therapeutics, such as Herceptin (*Albini et al., 2011*).

The ectodomains of all HER receptors comprise of four domains (I-IV), which in the ligand-free state in EGFR, HER3, and HER4 adopt a tethered conformation around a beta hairpin protrusion known as the dimerization arm (*Ferguson et al., 2003*; *Bouyain et al., 2005*). Early crystal structures of HER4/NRG1β, EGFR/EGF and EGFR/TGFα ectodomain (ECD) homodimers revealed that ligand binding between extracellular domains I and III causes a substantial conformational change that exposes the dimerization arm in domain II, allowing for the formation of active dimers stabilized through dimerization arm exchange between the monomers (*Figure 1a*; *Garrett et al., 2002*; *Ferguson, 2008*; *Liu et al., 2012*; *Ogiso et al., 2002*). In these symmetric ectodomain dimers, most of the interaction surface between the two receptors falls within the dimerization arm regions.

The orphan HER2 adopts an extended conformation in its apo state, thus being dimerization-competent without ligand binding (*Tzahar et al., 1996*; *Cho et al., 2003*). However, HER2 does not form stable homodimers under physiological expression levels and relies on heterodimerization with another ligand-bound HER receptor for activation (*Tzahar et al., 1996*). The inability to efficiently homodimerize might be encoded in the non-optimal manner with which HER2 engages a dimerization arm of a partner receptor, as illustrated in the recent structures of the NRG1β-bound HER2/HER3 and EGF-bound HER2/EGFR ectodomain heterodimers (*Diwanji et al., 2021*; *Bai et al., 2023*). When complexed with NRG1β-bound HER3, HER2 fails to engage the HER3 dimerization arm leaving only the HER2 arm engaged at the dimer interface (*Diwanji et al., 2021*). The dispensability of the HER3 arm at the interface is corroborated by the observation that its deletion does not impact HER2/HER3 dimerization and signaling (*Diwanji et al., 2021*). In the EGF-bound EGFR/HER2 structure, the EGFR dimerization arm binds HER2 but in a non-canonical manner characterized by increased dynamics and interactions of the arm with HER2 domains II and III instead of domain II and I observed in most other HER ECD dimers. As in the HER2/HER3 complex, the dimerization arm of the HER2 partner (in this case EGFR) is not required for heterodimerization and activation (*Bai et al., 2023*).

Together, the HER2-containing heterodimer structures reveal a dynamic mode with which HER2 engages a dimerization arm from a partner receptor (*Diwanji et al., 2021*). These dynamics suggest that HER homodimers are more stable and might preferentially form over HER2-containing heterodimers or heterodimers in general. This is consistent with repeated findings in cells expressing EGFR and HER2 in which a strong preference for EGFR homodimerization is observed over heterodimerization

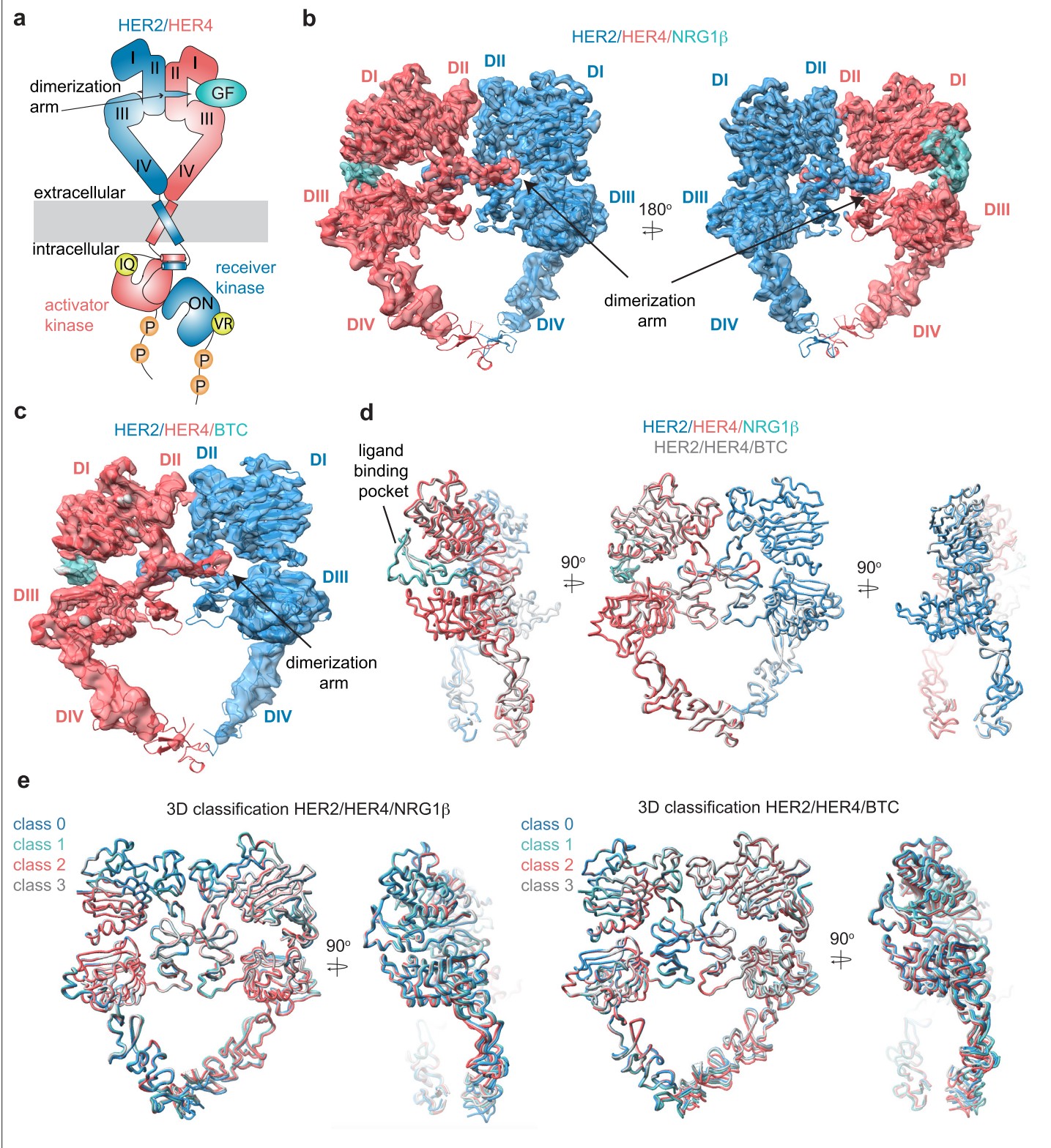

**Figure 1.** Structures of the HER2/HER4 heterodimers bound to NRG1β or BTC. (**a**) Cartoon schematic of the HER2/HER4/NRG1β heterodimer depicts the assembly of a 'heart-shaped' ectodomain dimer upon binding of a ligand/growth factor (GF) to HER4. Individual domains of the HER ectodomains are annotated as domain (**D**) I – IV. The intracellular kinase domains assemble into an asymmetric dimer in which HER2 adopts the receiver (activated) and HER4 the activator (inactive) positions, enforced by the interface mutations: HER2-V956R and HER4-I712Q, respectively. (**b–c**) Structures of the near full-length HER2-V956R/HER4-I712Q complex (labeled HER2/HER4) bound to NRG1β or BTC. The ectodomain models are shown in cartoon

*Figure 1 continued on next page*

*Figure 1 continued*

representation fitted into the cryo-EM density. Only density for the ectodomain modules was observed. Domains I-IV are labeled DI-DIV. (**d**) Overlay of HER2/HER4 heterodimers bound to NRG1β and BTC aligned on the HER2 chain (RMSD 0.835 Å). (**e**) 3D classification analysis of HER2/HER4 heterodimers bound to NRG1β or BTC. Overlay of models in ribbon resulting from the 3D classification of particles into four classes are shown (HER2/HER4/NRG1β 289,192 particles, HER2/HER4/BTC 148,541 particles). Models were aligned on the HER2 chain.

The online version of this article includes the following source data and figure supplement(s) for figure 1:

**Figure supplement 1.** Purification and the functional analysis of the HER2/HER4 heterodimers.

**Figure supplement 1—source data 1.** Original scan for Coomassie-stained gel in *Figure 1—figure supplement 1b*.

**Figure supplement 1—source data 2.** Original scan for Coomassie-stained gel in *Figure 1—figure supplement 1b* with all labels and cropped areas shown.

**Figure supplement 1—source data 3.** Original scan for Coomassie-stained gel in *Figure 1—figure supplement 1d*.

**Figure supplement 1—source data 4.** Original scan for Coomassie-stained gel in *Figure 1—figure supplement 1d* with all labels and cropped areas shown.

**Figure supplement 1—source data 5.** Original scan for western blot in *Figure 1—figure supplement 1f*.

**Figure supplement 1—source data 6.** Original scan for western blot in *Figure 1—figure supplement 1f* with all labels and cropped areas shown.

**Figure supplement 2.** Cryo-EM density maps of HER2/HER4 bound to NRG1β.

**Figure supplement 3.** Cryo-EM density maps of HER2/HER4 bound to BTC.

**Figure supplement 4.** Processing workflow for the HER2/HER4/NRG1β structure.

**Figure supplement 5.** Processing workflow for the HER2/HER4/BTC structure.

**Figure supplement 6.** Comparison between the HER2 and HER4 homo- and heterodimeric ectodomain structures.

with HER2 upon EGF treatment (*Bai et al., 2023*; *van Lengerich et al., 2017*; *Zhang et al., 2009*). Further evidence comes from biophysical studies in which isolated HER ectodomains were shown to form strong homodimers and only weakly detectable heterodimers in the presence of their cognate growth factors (*Ferguson et al., 2000*). However, this is not always the case and EGFR was reported to form heterodimers more favorably when stimulated with another ligand, betacellulin (BTC) (*Rush et al., 2018*). In addition, HER4 seems to engage equally as homodimers or as heterodimers with HER2, at least when interactions between isolated receptor ectodomains were measured (*Ferguson et al., 2000*). These receptor and dimer-specific idiosyncrasies highlight the importance of investigating HER receptor complexes with different ligands and a particular need for understanding how HER2 engages with HER4 – the final structure missing among HER2-containing HER complexes.

HER4 is activated by a diverse set of growth factor ligands including the neuregulin 1–4 family (NRG1-4), amphiregulin, epiregulin (EREG), and BTC (*Plowman et al., 1993*; *Sweeney et al., 2000*; *Trenker and Jura, 2020*). These ligands differ widely in their tissue expression and biological function (*Falls, 2003*; *Dunbar and Goddard, 2000*). For example, NRG1β plays essential roles in the development and functioning of the cardiovascular system and nervous system (*Falls, 2003*; *Meyer and Birchmeier, 1995*), while BTC is implicated in the differentiation of pancreatic β-cells (*Dunbar and Goddard, 2000*). Even in the same cells, these ligands induce distinct signaling outputs. In the human T lymphoblastic CEM cells stably expressing HER4, NRG1β (and NRG2β) are the most potent activators of AKT signaling, while BTC induces the strongest activation of ERK1/2 (*Sweeney et al., 2000*). These differential effects are likely due to a combination of factors. First, ligands are cross-reactive: NRG1βis also a ligand for HER3 while BTC also binds to EGFR (*Falls, 2003*; *Dunbar and Goddard, 2000*). Second, they might form structurally different HER4 ectodomain dimers, which in turn will affect dimer stability and downstream signaling, as observed for EGFR and its different cognate ligands (*Sweeney et al., 2000*; *Freed et al., 2017*). Third, the ligands might differentially modulate the degree of HER4 heterodimerization versus homodimerization (*Rush et al., 2018*; *Beerli and Hynes, 1996*). In particular, BTC appears to be uniquely poised to promote signaling by a wide range of HER heterodimers, including HER2/HER3 (*Dunbar and Goddard, 2000*; *Graus-Porta et al., 1997*; *Alimandi et al., 1997*), EGFR/HER3 and HER2/HER4 (*Rush et al., 2018*; *Dunbar and Goddard, 2000*; *Graus-Porta et al., 1997*; *Alimandi et al., 1997*). The mechanism for BTC-based dimerization of HER receptors remains unknown without structures of their complexes.

Whether HER4 dimers adopt different conformations while bound to different ligands, as seen in EGFR, has remained an open question as only crystal structures of NRG1β-bound HER4 dimers have

been reported (*Liu et al., 2012*). A spectrum of ligand-bound EGFR structures, including high-affinity (EGF and TGFα) or low-affinity (EREG), revealed different dimerization interfaces and underscored that the dimerization arm plays an important role in communication between the ligand binding pocket and EGFR dimer interface (*Freed et al., 2017*; *Huang et al., 2021*). In this study, we investigated these relationships for HER4, and its complexes with HER2. We focused on the comparison of NRG1β with BTC due to lack of structural insights into interactions of BTC with HER receptors, and its documented aptitude for promoting receptor heterodimerization in contrast to other ligands. Both ligands are known to promote HER4-dependent activation of HER2 (*Dunbar and Goddard, 2000*; *Graus-Porta et al., 1997*; *Alimandi et al., 1997*). We used cryo-electron microscopy (cryo-EM) to determine the first high-resolution structures of the NRG1β− and BTC-bound HER4/HER2 and HER4/HER4 ectodomain dimers in a full-length receptor context. Our analysis shows that there are no major differences between NRG1β− and BTC-bound complexes, but surprisingly in each case HER4 homodimers displayed large-scale dynamics compared with HER2/HER4 heterodimers. We also show that glycan modifications within the HER4 ectodomain extensively contribute to the HER4 homodimer interface, a feature previously not recognized in any other HER receptor complexes.

## Results

### Purification of the NRG1β- or BTC-bound HER2/HER4 heterodimers

To reconstitute the active HER2/HER4 complex for high-resolution structural analysis by cryo-EM, we introduced a G778D mutation in HER2 that prevents Hsp90 binding to the HER2 kinase domain and promotes HER2 heterodimerization as previously described (*Diwanji et al., 2021*; *Xu et al., 2005*; *Citri et al., 2004*). Both HER2 and HER4 receptors were truncated to remove their long, presumably unstructured, tails located C-terminal to the kinase domains and were transiently expressed in Expi293F cells individually. HER2 was expressed in the presence of canertinib, a covalent type-I kinase inhibitor that stabilizes the active conformation of the HER2 kinase. Cell lysates were pooled and in the first purification step, HER4 was affinity-purified via FLAG-tagged NRG1β or BTC (*Diwanji et al., 2021*; *Trenker et al., 2022*). In the second step, growth factor-bound HER4 complexes that interact with HER2 were enriched via a HER2-specific MBP-tag using amylose affinity resin (*Figure 1— figure supplement 1a* + 1b). Eluted proteins were further purified by size exclusion chromatography (*Figure 1—figure supplement 1c*) and dimeric fractions were frozen on graphene-oxide coated (GO) grids for cryo-EM analysis (*Figure 1—figure supplement 1e*, *Figure 1—figure supplement 2*, *Figure 1—figure supplement 3*).

We observed that HER2/HER4 dimers constituted only a small fraction of complexes purified using both ligands, indicating that in each case HER4 favored self-association (*Figure 1—figure supplement 1b* + 1d). HER receptor kinases asymmetrically dimerize in an active receptor complex, with one kinase adopting the function of an allosteric activator of the second kinase (receiver) (*Zhang et al., 2006*). To increase the yield of HER2/HER4 heterodimers vs HER4 homodimers, we introduced specific mutations that render the kinases activator only (N-lobe IQ mutation) or receiver only (C-lobe VR mutation). These mutations disrupt kinase homodimers but do not interfere with heterodimers in which the N-lobe mutant combined with the C-lobe mutant reconstitutes the asymmetric dimer (*Zhang et al., 2006*).

By introducing the relevant mutations, we designed HER2 and HER4 mutants to be compatible with two opposite activator/receiver configurations: HER4 activator (IQ)/HER2 receiver (VR) and HER2 activator (IQ)/HER4 receiver (VR) (*Figure 1a*). We first tested the signaling competency of these combinations, by transiently transfecting full-length HER2 and HER4 carrying respective mutations in COS7 cells and assessing receptor phosphorylation upon growth factor stimulation by Western blot analysis of cell lysates (*Figure 1—figure supplement 1e*). The HER2 constructs in these experiments do not feature the G778D mutation present in the constructs used for structure determination. Strikingly, the active heterodimer was only reconstituted in the HER4 activator (IQ)/ HER2 receiver (VR) configuration pointing to stereotyped roles that these two receptors play in the active complex irrespective of the activating growth factor (*Figure 1a*, *Figure 1—figure supplement 1f*). We used this set of interface mutations to enrich for the fraction of functional HER2-G778D-V956R/HER4-I712Q heterodimers in large-scale purification for structural studies. We will refer to these complexes simply as HER2/HER4.

# Cryo-EM structures of the HER2/HER4 heterodimers bound to NRG1β or BTC

We acquired cryo-EM datasets of the dodecyl-beta-maltoside (DDM)-solubilized, nearly full-length HER2/HER4 complexes with NRG1β or BTC on GO-coated grids (*Wang et al., 2020*). As reported in all previously published cryo-EM reconstructions of other RTKs (*Diwanji et al., 2021*; *Huang et al., 2021*; *Nielsen et al., 2022*; *Li et al., 2019*; *Uchikawa et al., 2019*; *Krimmer et al., 2023*), the cryo-EM density was the strongest in the ectodomain region of the receptor complex and the weakest within the transmembrane and intracellular domains (*Figure 1b–c*, *Figure 1—figure supplement 2a, f*, *Figure 1—figure supplement 3a, f*). Focused data processing on the ectodomains resulted in

**Table 1.** Cryo-EM data collection, refinement, and validation statistics.

| | HER2/HER4/ NRG1β (EMDB: EMD-41886) (PDB: 8U4L) | HER2/HER4/ BTC (EMDB: EMD-41885) (PDB: 8U4K) | HER4/ NRG1β (EMDB: EMD-41883) (PDB: 8U4I) | HER4/ BTC (EMDB: EMD:41884) (PDB: 8U4J) |
|---|---|---|---|---|
| **Data collection and processing** | | | | |
| Magnification | 105000 x | 105000 x | 105000 x | 105000 x |
| Voltage (kV) | 300 | 300 | 300 | 300 |
| Electron exposure (e–/Å$^2$) | 45.8 | 45.8 | 68.7 | 45.8 |
| Defocus range (μm) | 0.9–2.0 | 0.9–2.0 | 0.9–2.0 | 0.9–2.0 |
| Pixel size (Å) | 0.835 | 0.835 | 0.835 | 0.835 |
| Symmetry imposed | C1 | C1 | C1 | C1 |
| Initial particle images (no.) | 2938077 | 2261526 | 1264991 | 1715894 |
| Final particle images (no.) | 289192 | 148541 | 205726 | 274540 |
| Map resolution (Å) FSC threshold | 3.3 0.143 | 4.3 0.143 | 3.4 0.143 | 3.7 0.143 |
| Map resolution range (Å) | 3–7 | 3–7 | 3–7 | 3–7 |
| **Refinement** | | | | |
| Initial model used (PDB code) | 7MN5, 3U7U | 7MN5, 3U7U, AF-P35070-F1 | 3U7U | 3U7U, AF-P35070-F1 |
| Model resolution (Å) FSC threshold | 3.5 0.5 | 4.4 0.5 | 3.6 0.5 | 3.9 0.5 |
| Map sharpening *B* factor (Å$^2$) | –85 | –162.4 | –105.5 | –135.8 |
| Model composition Non-hydrogen atoms Protein residues Ligands | 9989 1230 35 | 9989 1228 35 | 10842 1301 57 | 10851 1297 58 |
| *B* factors (Å$^2$) Protein Ligand | 70.49 150.01 | 239.03 398.81 | 70.81 140.13 | 166.67 330.96 |
| R.m.s. deviations Bond lengths (Å) Bond angles (°) | 0.012 1.614 | 0.012 1.697 | 0.012 1.635 | 0.012 1.598 |
| Validation MolProbity score Clash score Poor rotamers (%) | 0.78 0.93 0.09 | 1.04 0.93 0.19 | 0.71 0.62 0.35 | 0.81 0.81 0.17 |
| Ramachandran plot Favored (%) Allowed (%) Disallowed (%) | 98.02 1.98 0 | 96.20 3.80 0 | 97.97 2.03 0 | 97.74 2.26 0 |

reconstructions of the NRG1β- and BTC-bound HER2/HER4 heterodimers at 3.3 Å and 4.3 Å resolution, respectively (*Figure 1b–c*, *Figure 1—figure supplement 4* and *Figure 1—figure supplement 5*, *Table 1*). In both structures, the ectodomains adopt a characteristic 'heart-shaped' arrangement observed in previously solved X-ray and cryo-EM structures of liganded HER receptor dimers (*Garrett et al., 2002*; *Liu et al., 2012*; *Ogiso et al., 2002*; *Diwanji et al., 2021*; *Bai et al., 2023*; *Freed et al., 2017*; *Huang et al., 2021*; *Lu et al., 2010*).

The HER2/HER4/NRG1β structures complete the panel of recently reported HER2 heterodimeric complexes. As in the HER2/HER3/NRG1β, HER2/EGFR/EGF, and HER2/EGFR/EREG structures (*Diwanji et al., 2021*; *Bai et al., 2023*), in the HER2/HER4 complexes the ligand-free HER2 enforces an asymmetric geometry within the heart-shaped ectodomain complex (*Figure 1b–c*, *Figure 1—figure supplement 6a*). The conformation adopted by HER2 is nearly identical in all complexes (*Figure 1—figure supplement 6b*, pairwise RMSD within HER2s 1.1–1.6 Å across different complexes), and is the same as observed in structures of an isolated HER2 ectodomain alone or in complex with therapeutic antibodies with only minor variations at the tip of the dimerization arm (*Figure 1—figure supplement 6b*). Thus, our structures are consistent with previous findings that HER2 does not undergo observable conformational changes upon heterodimerization with other HER receptors. The conformation adopted by HER4 in our structure is identical to a previously observed conformation in the crystal structure of isolated HER4 extracellular domain bound to NRG1β (RMSD 1.7 Å – 4.4 Å across different structures). This conformation also closely matches the extended state of HER3 and EGFR in their respective heterodimers with HER2 (RMSD 2.2 Å, RMSD 2.6 Å, respectively) (*Figure 1—figure supplement 6b*; *Diwanji et al., 2021*; *Bai et al., 2023*).

Despite the diverse sequences of the NRG1β and BTC ligands, the larger-scale domain conformation of the HER2/HER4 heterodimers stabilized by each ligand is identical with only small differences in the ligand binding pockets (*Figure 1d*). Due to the lower resolution of the HER2/HER4/BTC complex, we cannot exclude the possibility of differences in side-chain arrangements between the two structures. However, we attribute the lower resolution to variability in data collection on GO grids, rather than differences in conformational heterogeneity of HER2/HER4/BTC. Recently published structures of EGFR homodimers induced by binding of the two high-affinity ligands, EGF, and TGFα, revealed that binding of these two different ligands results in distinct ensembles of EGFR dimer conformations, specifically within domains IV, seemingly coupled to scissor-like movements around the dimerization arm region (*Huang et al., 2021*). To investigate whether NRG1β and BTC might lead to similar effects in the HER2/HER4 structures, we have performed an equivalent analysis. Extensive 3D variability and 3D classification analysis of the HER2/HER4/NRG1β and HER2/HER4/BTC datasets did not reveal any defined conformational heterogeneity within domains IV or dimerization arm regions (*Figure 1e*).

## Differences between HER2-containing heterodimers

Like other HER receptor dimers, HER2 and HER4 heterodimerize mainly via interactions between domains II, with significant contributions from the dimerization arms (*Figure 2a–b*), which carry conserved sequence features across HER family (*Figure 2—figure supplement 1a*). The total buried surface area (BSA) at the HER2/HER4 heterodimer interface (domains I-III, measured using UCSF ChimeraX) encompasses 2784 Å$^2$ and is comparable to HER4 and EGFR homodimers (2768–2866 A$^2$, PDB: 3U7U; 3006 A$^2$, PDB: 3NJP, respectively) and the EGFR/HER2 heterodimer (2864 A$^2$, PDB: 8HGO), while the HER2/HER3 heterodimer interface is significantly smaller (2066 A$^2$, PDB: 7MN5) (*Figure 2—figure supplement 1b*). Additional stabilization comes from the interface region above the dimerization arms, which in the HER2/HER4 heterodimer is predominantly stabilized by polar interactions, with few hydrophobic Van-der-Waals contacts (*Figure 2b*, *Figure 2—figure supplement 1b*). The HER2/HER4 interface has two salt bridges (HER2 H237 – HER4 D218 and HER2 E265 – HER4 R232) compared to only one in HER2/EGFR and none in HER2/HER3 (*Figure 2b* box A).

Given the non-canonical engagement of the dimerization arm of the HER2 partner receptor in previously solved HER2-containing heterodimer structures, we analyzed the HER4 dimerization arm in our structures. HER2 and HER4 dimerization arms are resolved in both NRG1β and BTC-bound HER2/HER4 complexes (*Figure 1b–c*, *Figure 2a*). The HER2 dimerization arm is stabilized by several polar and Van-der-Waals interactions with the dimerization arm-binding pocket of HER4, which involve two conserved aromatic residues, specifically Y274 and F279 in HER2 that interact with HER4 G286, C304, and R306 (*Figure 2b* box B). The equivalent residues in the HER4 dimerization arm, Y268 and F273,

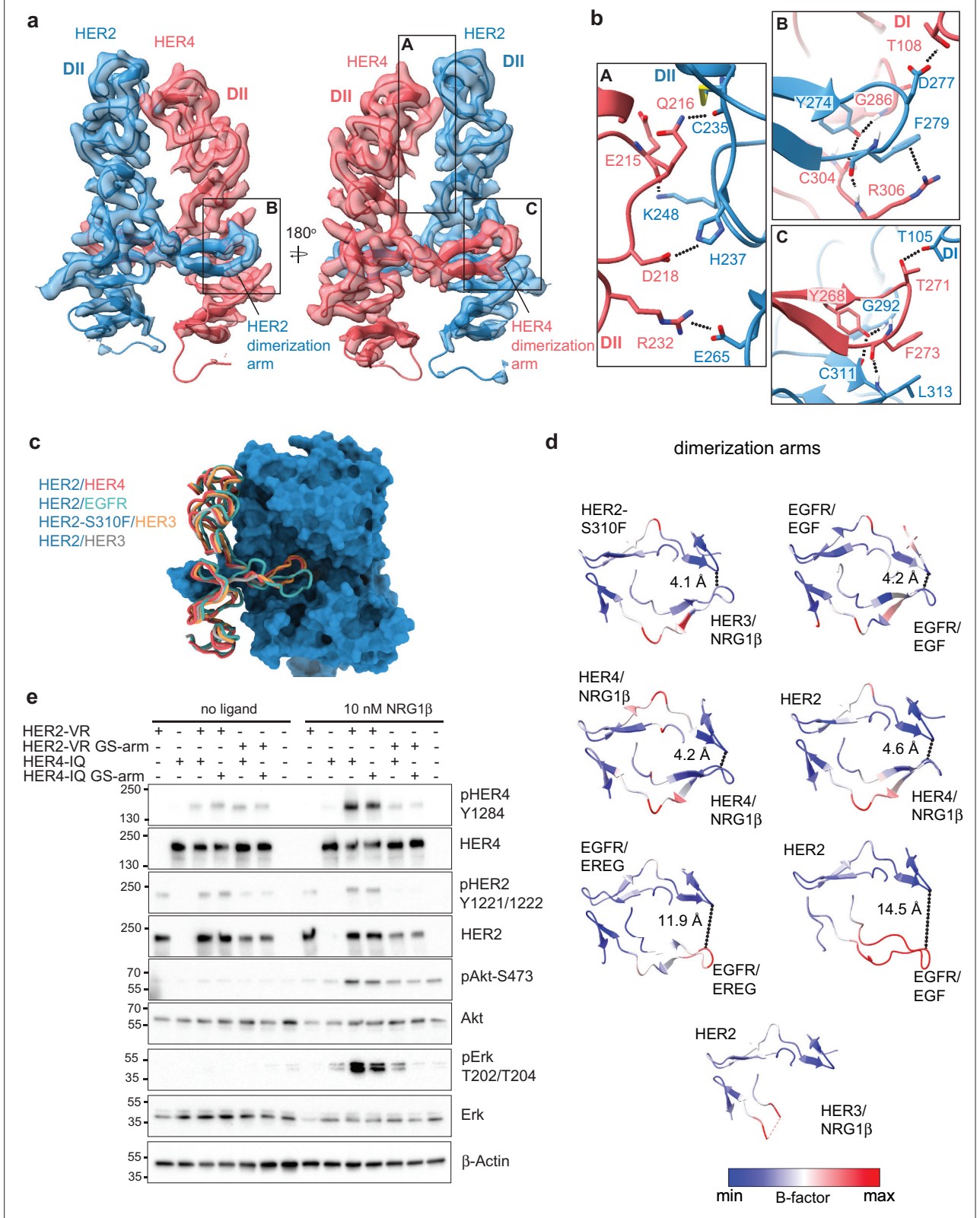

**Figure 2.** Near symmetric engagement of the HER2 and HER4 dimerization arms at the dimerization interface. (**a**) Cryo-EM density and model of the HER2/HER4/NRG1β domain II at two different orientations highlight two equally well-resolved dimerization arms. (**b**) Hydrogen-bonds, cation-π interactions, and salt bridges are depicted at the dimer interface, with other residues omitted for clarity. The HER2 and HER4 dimerization arms engage in the same set of polar interactions (insets B and C), except for a cation-π interaction between HER2 F279 with HER4 R306 (**A**) due to a substitution of

*Figure 2 continued on next page*

*Figure 2 continued*

the equivalent of HER2 R306 to L313 in HER2. Residues labeled 'DI' are in receptor domain I while all others are in domain II (DII). Interface residues and hydrogen bonds were determined using UCSF ChimeraX. (**c**) Known HER2 heterodimers are aligned using the HER2 chain to highlight the positioning of the dimerization arms. (**d**) Dimerization arm regions of selected HER receptor dimers are shown colored by B-factors. B-factor colors were scaled to represent max and min B-factor values within each structure corresponding to different absolute values across structures due to variability in their resolution. Distance measurements at fixed points highlight a correlation between asymmetrically distributed B-factors and asymmetrically engaged dimerization arms. (**e**) Western blot analysis of NR6 cell lysates transduced with indicated HER2 and HER4 constructs. Cells were starved for 4 h prior to stimulation with 10 nM NRG1β at 37 °C for 10 min. Molecular weight markers (in kDa) are indicated next to each blot.

The online version of this article includes the following source data and figure supplement(s) for figure 2:

**Source data 1.** Original files for western blot analysis in *Figure 2e*.

**Source data 2.** Original files for western blot analysis in *Figure 2e* with all labels and cropped areas shown.

**Figure supplement 1.** Detailed view of the dimer interfaces of HER receptor homo- and heterodimers.

are engaged in reciprocal interactions with the backbone atoms of HER2 G292, C311, and L313 via a network of hydrogen bonds (*Figure 2b* box C). These aromatic dimerization arm residues are strictly conserved as phenylalanine or tyrosine residues in all HER receptors (*Figure 2—figure supplement 1a*) and participate in the same interactions in the structures of the symmetric EGFR/EGF and HER4/NRG1β ectodomain homodimers (*Figure 2—figure supplement 1b*). In the EGFR/HER2 and HER3/HER2 heterodimers, only the HER2 dimerization arm makes these interactions (*Figure 2—figure supplement 1b*). The EGFR dimerization arm is rotated out of the canonical dimerization arm binding pocket of HER2, preventing such interactions, and the dimerization arm of HER3 is not even resolved (*Figure 2c–d*, *Figure 2—figure supplement 1b*; *Garrett et al., 2002*; *Liu et al., 2012*). Thus, in this regard, HER2/HER4 heterodimers are more similar to known structures of HER homodimers (NRG1β-bound HER4 and EGF-bound EGFR homodimers) than to heterodimers.

These similarities are also reflected in the interactions that the tips of both dimerization arms at the HER2/HER4 interface make with domains I of their respective dimerization partners. The D277 in the HER2 tip hydrogen bonds with T108 in domain I of HER4, while the T271 in the HER4 tip hydrogen bonds with T105 in HER2 domain I (*Figure 2b*). The HER2/HER4 complex is the only HER2 heterodimer that involves domain I of both receptors in the dimer interface via the tip of the dimerization arms in a near-symmetric fashion (*Figure 2d*, *Figure 2—figure supplement 1b*). Other HER2 heterodimers and, incidentally also the EGFR/EREG dimer, exhibit an asymmetric dimerization arm configuration with one dimerization arm being less engaged, evidenced by increased B-factors (*Figure 2d*). Thus, the relative orientation of two HER monomers varies among all HER heterodimer structures (*Figure 1—figure supplement 6c*). EGFR and HER3 exhibit a hinging motion in the direction of the HER2 dimerization arm in comparison to HER4, which is rotated slightly away from it (*Figure 1—figure supplement 6c*). Interestingly, the only other instance when a HER2-containing heterodimer is observed to make symmetric interactions is when HER2 carries an oncogenic mutation, S310F, in the heterodimeric complexes with HER3 (*Figure 1—figure supplement 6c*; *Diwanji et al., 2021*). This suggests that HER2/HER4 heterodimers are the most stable among HER2 heterodimers. Consistent with this notion, previous studies showed that the recombinant HER2 and HER4 ECDs form the most stable heterocomplex among all other HER heterodimers, with efficiency similar to HER4 homodimers (*Ferguson et al., 2000*).

## The dimerization arm of HER2, but not HER4, is required for HER2/HER4 activation

The EGFR and HER3 dimerization arms are dispensable for signaling within their respective heterodimers with HER2, a property attributed to their high flexibility and disengagement from the HER2 dimerization arm binding pocket (*Diwanji et al., 2021*; *Bai et al., 2023*). The canonical binding mode of the HER4 dimerization arm in the HER2/HER4 dimer structures raises the question whether it is required for signaling by this complex. To test the role of HER4 arm, we transduced full-length HER2-VR (V956R) and HER4-IQ (I712Q) constructs into murine NR6 cells. Dimerization arm sequences were replaced either in HER2 or HER4 with a flexible loop of alternating glycine and serine residues as previously described (GS-arm) (*Diwanji et al., 2021*; *Bai et al., 2023*). NRG1β-induced phosphorylation of HER2, HER4, ERK, and AKT was not notably affected by the substitution of the HER4

dimerization arm to a GS-arm relative to receptors featuring wild-type dimerization arm sequences, indicating that the HER4 dimerization arm is not required for assembly and activation of HER2/HER4 heterodimers (*Figure 2e*). In contrast, the substitution of the HER2 dimerization arm sequence fully abolished the activation of the heterocomplex, as previously reported (*Figure 2e*; *Bai et al., 2023*). Small increases in pERK levels in cells expressing the HER4-IQ construct are consistent with previous observations that the IQ mutation in HER kinase domains has small residual activity through homodimerization (*Zhang et al., 2006*). Thus, despite full engagement at the interface of both dimerization arms in the HER2/HER4 complexes, the HER4 arm is still dispensable for activation, and it is the HER2 arm that potentiates the formation of the active complex.

## HER4 homodimers display higher large-scale conformational flexibility than HER2/HER4 heterodimers

Our cryo-EM structures of the full-length HER2/HER4 complexes bound to either NRG1β or BTC, did not reveal discernible differences at the receptor dimerization interface and larger-scale domain arrangements (*Figure 1d*). In other cryo-EM structures of the full-length HER2 heterodimers, EGFR/HER2 bound to a high-affinity EGFR ligand, EGF, or a low-affinity EREG, any differences are also imperceptible (*Bai et al., 2023*). Ligand-specific differentiation of structural states becomes only evident in EGFR homodimers. The most drastic example is the breaking of C2 symmetry in the crystal structures of EGFR ectodomain homodimers bound to EREG vs symmetric structures of EGFR with EGF or TGFα (see *Figure 1—figure supplement 6a* for symmetry axes in HER receptor dimers) (*Freed et al., 2017*). However, even in the symmetric crystal structures of EGFR bound to two high-affinity ligands, EGF and TGFα, there are differences in intermonomer EGFR angles between the two ligand complexes (*Garrett et al., 2002*; *Ogiso et al., 2002*). As mentioned above, cryo-EM analysis of the full-length EGFR homodimers, extensive 3D classification, and variability analysis revealed that both EGF and TGFα stabilize a range of EGFR dimer shapes with different intermonomer angles, but they differ in their ability to stabilize conformations with large intermonomer angles in which membrane-proximal domains IV are separated (*Huang et al., 2021*).

These comparisons raise the question of whether HER homodimers explore a wider range of conformations compared to heterodimers, specifically those singly-liganded heterodimers that contain the orphan HER2 receptor. To test this hypothesis for HER4 complexes, we determined the cryo-EM structures of full-length HER4 homodimers bound to NRG1β or BTC (*Figure 3a*, *Figure 3—figure supplement 1*, *Figure 3—figure supplement 2*, *Figure 3—figure supplement 3*). In both structures, HER4 kinase domains were bound to afatinib to enable high-resolution reconstruction in the ECD module (3.4 Å for NRG1β, and 3.7 Å for BTC) (*Figure 3—figure supplement 2*, *Figure 3—figure supplement 3*). AMP-PNP/Mg$^{2+}$-bound or apo HER4/NRG1β complexes resulted in a similar overall reconstruction, albeit at lower resolution (4.2 Å and 3.9 Å, respectively) (*Figure 3—figure supplement 1f–g*). As observed in a previous HER4/NRG1β crystal structure of isolated ECDs, liganded HER4 assembles into homodimers with near-perfect C2 symmetry (*Figure 3a*, *Figure 3—figure supplement 4*). However, while we observed that applying C2 symmetry in the final refinement step nominally improved resolution (*Figure 3—figure supplement 5*, *Figure 3—figure supplement 6*), closer inspection of our final reconstructions in the absence of applied symmetry suggests our structures, similar to other published structures of HER homodimers, are not perfectly symmetric (*Figure 3—figure supplement 4*). Thus, we focused our analysis on reconstructions without enforced symmetry.

The HER4/NRG1β and HER4/BTC homodimers adopt overall similar conformations, but our reconstructions show differences in the intermonomer angles that are adopted by NRG1β- vs BTC-bound homodimers (*Figure 3b*). The two receptors in the HER4/NRG1β dimer adopt an angle of 37 degrees compared to 33 degrees for HER4/BTC. Detailed 3D classification analysis revealed substantial scissor-like movements around the dimerization arm in both data sets with the concerted intermonomer movement of domains I and IV, varying from 35 to 39 degrees for HER4/NRG1β and 30–35 degrees for HER4/BTC (*Figure 3c*). These differences are not as pronounced as observed between 'separated' and 'juxtaposed' states of EGFR domain IV in EGF vs TGFα bound EGFR homodimers (*Huang et al., 2021*) but persisted through multiple different processing methodologies (see methods). Such observations are indicative of the increased dynamics present in homodimeric HER4 receptor assemblies compared to their HER2-bound heterodimer counterparts. This difference in intermonomer angles was maintained even with C2 symmetry applied to the final refinement steps and 3D classification.

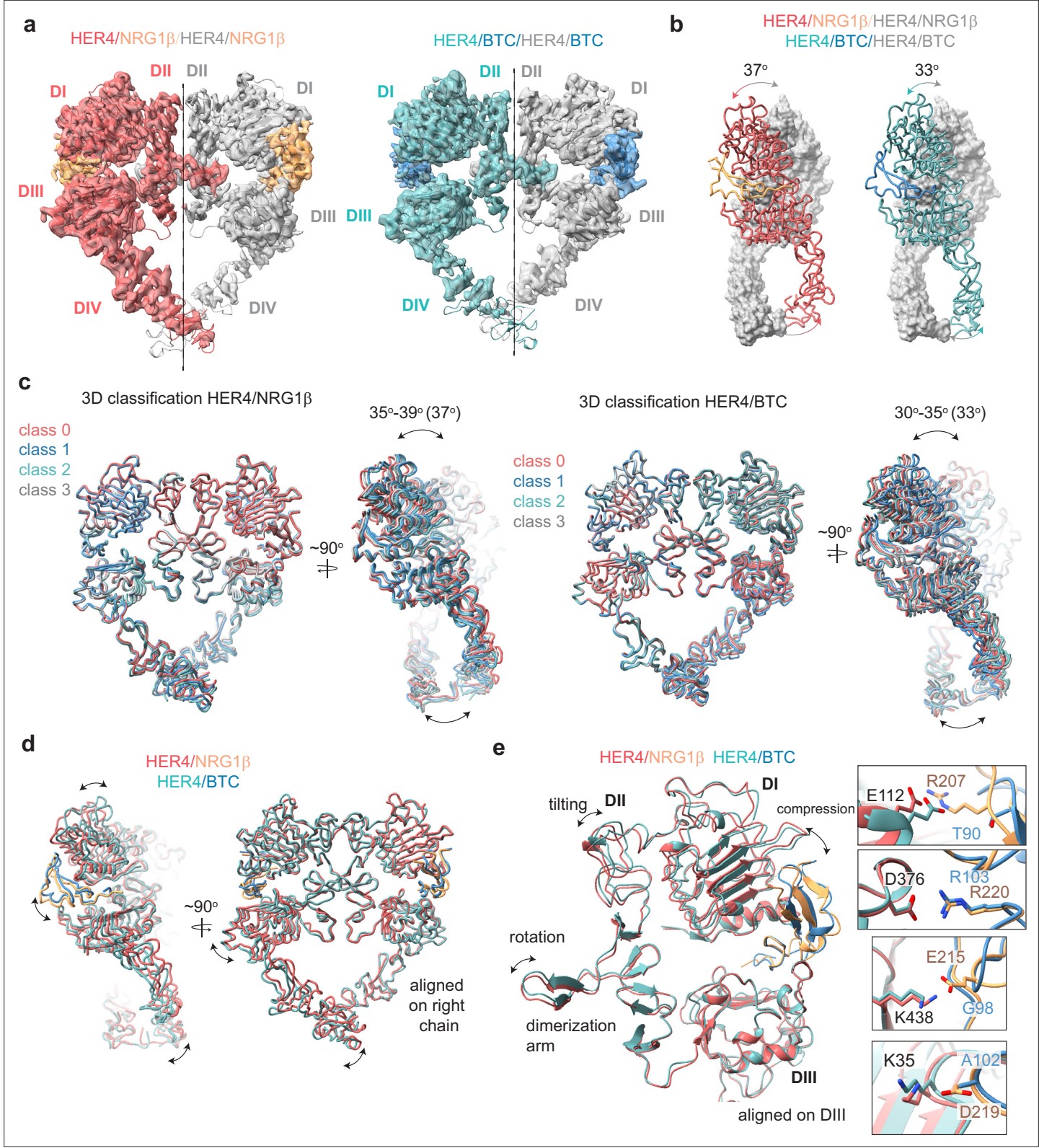

**Figure 3.** Structures of HER4 homodimers bound to NRG1β or BTC reveal ligand-specific conformational heterogeneity. (**a**) Structures of full-length HER4 homodimers bound to either NRG1β or BTC. Only density for the ectodomain modules was observed in both structures, shown here as a cartoon representation fitted into the cryo-EM density. (**b**) Comparison between the NRG1β− and BTC-bound HER4 dimers. Angle measurements were derived using UCSF ChimeraX by defining an axis through each receptor in a dimer and measuring the angle between the two axes. (**c**) Overlays of ribbon

*Figure 3 continued on next page*

*Figure 3 continued*

models obtained by 3D classification of particles into four distinct classes are shown for HER4 homodimers bound to NRG1β or BTC (205,726 particles HER4/NRG1β and ~274,540 particles HER4/BTC). Classification was performed in cryoSPARC using the heterogeneous refinement job with four identical start volumes and particles from final reconstructions are shown in (**a**). (**d-e**) Overlays of HER4 receptor homodimers bound to NRG1β or BTC show differences in the ligand binding pockets and how receptors assemble into dimers. Receptors were aligned as indicated in the panels. The HER4-NRG1β engages 4 salt bridges in the binding pocket, three of which are not present in HER4-BTC (shown in boxes). The salt bridge involving HER4 K35 can only be confidently observed in cryo-EM maps of one monomer (chain A).

The online version of this article includes the following source data and figure supplement(s) for figure 3:

**Figure supplement 1.** Purification of HER4 homodimers bound to NRG1β or BTC and structural analysis.

**Figure supplement 1—source data 1.** Original scans for Coomassie-stained gels in *Figure 3—figure supplement 1b*.

**Figure supplement 1—source data 2.** Original scans for Coomassie-stained gels in *Figure 3—figure supplement 1b* with all labels and cropped areas shown.

**Figure supplement 2.** Cryo-EM density maps of HER4 bound to NRG1β processed without symmetry applied.

**Figure supplement 3.** Cryo-EM density maps of HER4 bound to BTC processed without symmetry applied.

**Figure supplement 4.** HER4 homodimers do not show ideal C2 symmetry.

**Figure supplement 5.** Processing workflow and data statistics for the HER4/NRG1β homodimer.

**Figure supplement 6.** Processing workflow and data statistics for the HER4/BTC homodimer.

The structural origins of the differences in intermonomer angles within the NRG1β and BTC-bound HER4 homodimers are challenging to explain. Due to the differences in intermonomer movement, overlays between NRG1β and BTC-bound HER4 differ slightly, however overall, the two homodimers are almost identical (*Figure 3d–e*). Nevertheless, there are unique residue interactions within the ligand-binding pockets that correlate with changes in dimerization arm positioning, which might ultimately dictate the angle at which the two receptors engage with one another (*Figure 3d–e*). While both pockets bury a similar surface area (NRG1β: 2967–3068 Å$^2$, BTC: 3142–3163 Å$^2$), the HER4/NRG1β pocket is characterized by a larger network of ionic interactions. In the HER4/NRG1β pocket, HER4 K438, K35, E112 and D376 form salt bridges with NRG1β E215, D219, R207 and R220, respectively. In the HER4/BTC pocket, HER4 D376 also engages an equivalent BTC R103, but the other three residues are substituted with hydrophobic/polar residues in BTC (NRG1β E215=BTC G98, NRG1β D219=BTC A102, NRG1β R207=BTC T90) (*Figure 3e*). Thus, substitutions into small, apolar residues in BTC result in a more 'compressed' ligand-binding pocket in HER4/BTC homodimers than in HER4/NRG1β homodimers, which may allosterically determine different intermonomer angles via modulation of dimerization arm positioning (*Figure 3b–e*).

## HER4 glycosylation reveals structural stabilization via glycans that bridge extracellular subdomains and receptor dimers

The conformation of the HER4/NRG1β cryo-EM homodimer deviates slightly from the three crystallographic HER4/NRG1β homodimers present in the asymmetric unit (PDB ID: 3U7U) in which each monomer adopts a different orientation of the domain IV relative to the rest of the ectodomain (*Figure 3—figure supplement 4a*, RMSD: 5.438 Å, 5.435 Å, and 3.662 Å). Notably, the two cryo-EM HER4 homodimer structures are more symmetric. RMSDs for monomers within the cryo-EM dimers are 1.42 Å in the HER4/NRG1β homodimer and 1.58 Å in the HER4/BTC homodimer (*Figure 3—figure supplement 4b*+c), as compared to the three crystallographic dimers in which the HER4 monomers align with RMSDs of 1.67 Å, 5.76 Å, and 2.38 Å (*Liu et al., 2012*). Several reasons could account for this variation, including consequences of crystal packing or lack of intracellular and transmembrane domains, which are present in our constructs, albeit not resolved in cryo-EM density. Another explanation is differences in HER4 glycosylation in our cryo-EM sample purified from human cells as compared to deglycosylated HER4 ectodomains used for crystallography, which only maintain the first N-acetyl-glucosamine (NAG) on asparagine residues that are N-glycosylated (*Liu et al., 2012*).

Our cryo-EM analysis of full-length HER4 homodimers reveals multiple well-resolved N-linked glycans in receptor ectodomains and points to their role in stabilizing interactions between two receptor monomers. HER4 features 11 known N-glycosylation sites (as defined in Uniprot (ID: Q15303): N138, N174, N181, N253, N358, N410, N473, N495, N548, N576, and N620). Eight of them are resolved in the cryo-EM maps of both HER4/NRG1β and HER4/BTC homodimers that enabled the building

of core glycan trees up to 5 sugar moieties (*Figure 4a*, *Figure 4—figure supplement 1*, shown for HER4 /NRG1β). While two glycans on HER4 domain III (N410-linked and N495-linked) point away from any interfaces in a receptor homodimer, the remaining six are positioned to mediate intra- or inter-domain contacts within the receptor dimer. The N253-linked glycan in domain II has a well-resolved density that appears to be continuous with a density coming from the N138-linked glycan in domain I. The trees up of five sugars can be reasonably placed into each density pointing towards direct glycan-glycan and glycan-protein interactions across HER4 sub-domains I and II (*Figure 4a* box A and B). Similarly, the N548-linked glycan on domain IV points toward the C-terminal end of the domain II and is poised to mediate an intramolecular glycan-protein contact between the two domains (*Figure 4a* box C).

Most remarkably, at lower contour levels, our HER4 maps show continuous cryo-EM density connecting the two receptor monomers originating from N548 and N576 on domain IV of one receptor monomer and N358 on domain III of the other receptor involving sugar moieties beyond the core glycan trees of 4–5 sugars (*Figure 4a* box D, *Figure 4—figure supplement 1a*). This points to a direct contribution of N-linked glycosylation towards the HER4 homodimer interface that, given the low resolution, is likely structurally heterogenous and involves complex glycosylation trees attached to the respective asparagine residues. Indeed, 3D classification of the particles in our final reconstruction uncovered at least one class with more defined N548-N576-N358 glycan networks in the dimer inter-face (*Figure 4—figure supplement 1b*). Thus, our analysis of the HER4/NRG1β homodimer cryo-EM density uncovers an important role of receptor glycosylation in stabilizing the active HER4 monomer by bridging its two distal domains, domain II and IV and domains I and II, and likely both monomers within the HER4 homodimer through direct inter-receptor connections.

N-linked glycosylation is also resolved in the cryo-EM maps of the HER2/HER4 heterodimers (NRG1β and BTC bound), with HER4 glycosylation patterns being the same as seen in homodimers. HER2 has seven N-linked glycosylation sites (as defined in Uniprot (ID: P04626): N68, N124, N187, N259, N530, N571, and N629), five of which are visible in the cryo-EM maps (N68, N187, N259, N530, and N517) (*Figure 4b*, shown for the HER2/HER4/NRG1β heterodimer). As in the case of HER4, some glycans on HER2 mediate direct interdomain contacts within HER2, similar to the ones previously observed in the crystal structure of HER2 with pertuzumab, albeit more sugar moieties can be built in our structure (*Franklin et al., 2004*). The first three sugar moieties on N259 in domain II are particularly well-resolved and appear to directly engage the domain I polypeptide chain (*Figure 4b* box B). However, in contrast to the HER4 homodimers, we do not observe continuous density connecting the two heterodimer monomers indicating that glycan-mediated interfaces seen in HER4 homodimers cannot be established in HER2/HER4 heterodimers. This is because HER2 does not have glycosylation consensus sites equivalent to HER4 N358, N548, and N576 (*Figure 4b* box C). Based on these observations, it is tempting to speculate that the higher propensity for HER4 to homodimerize rather than heterodimerize with HER2 observed in our pull-downs (*Figure 1—figure supplement 1b*) is at least partially rooted in the stabilization of the homodimer by glycan-mediated interactions.

## Discussion

### First structures of HER2/HER4 and BTC complexes

We present here the first cryo-EM reconstructions of both the homodimer and heterodimer complexes of the HER4 receptor in its full-length form, bound to two different high-affinity HER4 cognate growth factors, NRG1β and BTC. This is also the first time that a betacellulin growth factor has been resolved bound to a HER receptor. The HER2/HER4 heterodimer structures now complete the ensemble of possible HER receptor heterodimer structures that involve the orphan HER2 receptor. Only the ecto-domains are resolved in our structures, as repeatedly has been the case for any full-length receptor tyrosine kinase reconstructions (*Diwanji et al., 2021*; *Huang et al., 2021*; *Nielsen et al., 2022*; *Li et al., 2019*; *Uchikawa et al., 2019*; *Krimmer et al., 2023*). While not resolved, interactions contrib-uted by the intracellular domains appear to be essential for the stabilization of the receptor complexes in our cryo-EM reconstructions. In our previous analysis of the HER2/HER3/NRG1β complex, the introduction of oncogenic mutations in the HER3 pseudokinase that increase its dimerization affinity with HER2, and the presence of HER2 kinase inhibitors was essential for efficient heterodimer recon-stitution and improved resolution (*Diwanji et al., 2021*; *Trenker et al., 2022*). Similarly, for HER4

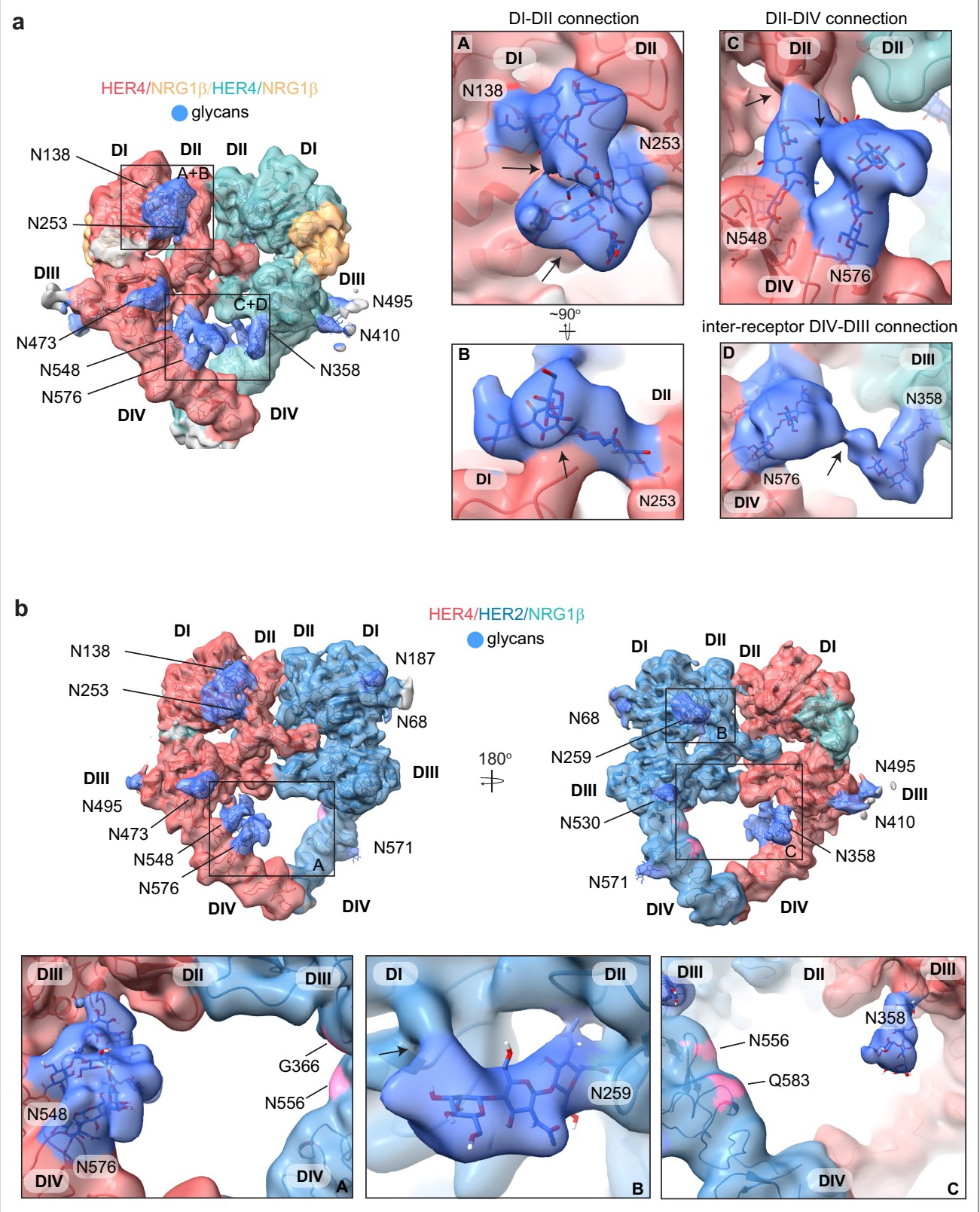

**Figure 4.** HER4 homodimers are stabilized via inter-receptor glycans. (**a**) Model of the HER4/NRG1β homodimer fitted into the cryo-EM density, lowpass-filtered to 6 Å, reveals multiple glycans that mediate intra- and interreceptor connections. Glycans are shown in blue. Insets (**A** and **B**) are close-up views of glycans connected to N138 and N253, and are shown at a higher volume contour than the central heterodimer. Insets (**C** and **D**) are close-up views of glycans connected to N548, N576, and N358. (**D**) shows continuous glycan density originating from N576 of one receptor and

*Figure 4 continued on next page*

*Figure 4 continued*

connecting to N358 of the dimerization partner. Maps are shown at lower contour than in the central heterodimer. Various contour levels are shown in *Figure 3—figure supplement 4a* for reference. Arrows indicate regions in which the cryo-EM map from one glycan merges with the density of glycans or polypeptide chains from different HER receptor sub-domains. (**b**) Model of HER2/HER4/NRG1β fitted into cryo-EM density, lowpass-filtered to 6 Å, reveals intra-receptor glycosylation only. Insets (**A**) shows HER4 glycosylation on N548 and N576 pointing from HER4 domain IV to domain II, but less pronounced as observed in HER4 homodimers. Glycan connections between domain I and II in HER4, via N138 and N253-linked glycans, are comparable to the ones seen in HER4 homodimer shown in inset (**A**). Inset (**B**) shows the equivalent glycan connections in domain I and II of HER2. Inset (**C**) reveals missing glycosylation sites at equivalent positions in HER2; G366, N556, and Q583 (pink).

The online version of this article includes the following figure supplement(s) for figure 4:

**Figure supplement 1.** Glycosylation in the cryo-EM structures of HER4/NRG1β homodimer and comparison to other HER receptors.

structures reported here, selective enrichment of kinase heterodimers via the introduction of activator/receiver mutations in HER2/HER4 and the introduction of kinase inhibitors to homo- and heterodimer complexes improved the resolution of cryo-EM reconstructions.

## Conserved features of HER2 heterodimers

Across the family, the three HER2-containing heterodimers adopt an asymmetric heart-shaped ecto-domain structure and in each one of them the HER2 conformation is identical while the dimerization interface is unique. The main difference centers on the engagement of the dimerization arm extended to HER2 by the partner receptors. The HER2/HER3 interface is most dynamic with the HER3 dimerization arm not being resolved at all (*Diwanji et al., 2021*). In the HER2/HER4 and the HER2/EGFR structures, HER4 and EGFR dimerization arms are resolved but make unique interactions with HER2 (*Bai et al., 2023*). The EGFR dimerization arm engages HER2 via non-canonical interactions with domain III that resemble those only observed in the crystal structure of the EGFR/EREG ectodomain complex (*Bai et al., 2023*; *Freed et al., 2017*). In comparison, the HER4 dimerization arm in the HER2/HER4 heterodimer presented here is engaged with HER2 via several canonical interactions, observed across most of the HER receptor homodimer structures (*Garrett et al., 2002*; *Liu et al., 2012*; *Ogiso et al., 2002*; *Lu et al., 2010*). This binding mode might explain HER2/HER4 heterodimer seems to be most stable among HER2-containing heterodimers, as measured by studying associations between isolated HER ectodomains (*Ferguson et al., 2000*).

While the positioning of the HER2 and HER4 dimerization arm appears almost symmetric, the number of hydrogen bonds formed by the HER4 dimerization arm is reduced compared to that of HER2. In addition, HER2 fails to engage its partner receptors via a conserved cation-π interaction that is exchanged by both monomers in all symmetric EGFR and HER4 homodimers, which in HER4 involves a dimerization arm phenylalanine (F273) and a domain II arginine (R306). The arginine is a leucine in HER2 (L313). It had been speculated previously that the inability of HER2 to form this interaction may be the reason for the non-canonical placement of HER3 and EGFR dimerization arms in their respective heterodimers with HER2 (*Diwanji et al., 2021*; *Bai et al., 2023*). However, our HER2/HER4 heterodimer structure shows that even without the cation-π interaction, the HER4 dimerization arm can be placed in a canonical position. Lastly, the overall weaker interactions that HER2 makes with the dimerization arms of its partner receptors are likely the reason why these partner arms are not needed for the stabilization of the active signaling HER2 heterodimers. This has been observed for the HER2/HER3 and HER2/EGFR complexes (*Diwanji et al., 2021*; *Bai et al., 2023*), and we show here that the same is true for the HER2/HER4 heterodimer.

## Homodimerization vs heterodimerization

The extent of HER receptors propensity to form homodimers versus heterodimers, and their functional significance, are topics of ongoing debate. Some ligands, like EGF, are well documented to favor homodimers of their cognate receptors (EGFR in this case), while others, like BTC, have been shown to more readily promote hetero-association (*Rush et al., 2018*; *Beerli and Hynes, 1996*). While in a cellular context, there might be many factors that shape these equilibria, including relative levels of receptor expression, their localization within membrane microdomains, and/or interaction with other, yet unknown, factors that might stabilize certain dimer combinations, our studies bring insights into these interactions in a simplified in vitro system. We note that both NRG1β and BTC favor HER4 homo-association, and only a small fraction of complexes purified using pulldowns with these

ligands immobilized on beads yielded HER2/HER4 heterodimers. This was the case even when HER2 and HER4 kinase domains carried mutations designed to prevent their homo-associations and to favor heterodimerization. This phenomenon has been previously observed for the EGFR/HER2 system, where co-expressed receptors stimulated with EGF formed almost exclusively EGFR homodimers upon detergent extraction, with the limited formation of EGFR/HER2 heterodimers (*Bai et al., 2023*). The same study also reported only a small fraction of heterodimerization (<10%) by live-cell single molecule imaging of EGFR and HER2 on the plasma membrane of EGFR/HER2 positive SUM159 cells after EGF stimulation. Altogether, these findings raise questions about the conditions under which HER receptor heterodimers form in vivo, especially for HER receptors, which are not obligate heterodimers, namely EGFR and HER4. Is their reluctance to form such heterodimers an important part of the regulation of their signaling specificity, a current lack of knowledge about the conditions under which they form, or both? For example, one of the consequences of HER2 overexpression in cancer could be the elevation of the otherwise non-optimal heterodimers with EGFR, resulting in potentiation of oncogenic signaling.

## Biased agonism

The degree of symmetry between ectodomains of EGFR in the active liganded dimer has been correlated with the strength of its signaling output. Ligand binding is allosterically coupled to the positioning of the dimerization arm and depends on how the ligand engages domains I and III. In EGFR, this allosteric path is differentially engaged by low-affinity EGFR ligands (EREG, Epigen) vs high-affinity ligands (EGF, TGFα), resulting in asymmetric and dynamic dimers vs symmetric and stable dimers, respectively (*Garrett et al., 2002*; *Ogiso et al., 2002*; *Freed et al., 2017*). The weak asymmetric EGFR dimers have been shown to correlate with more sustained activation of ERK and AKT pathways leading to differentiation, while the more stable symmetric dimers induce more transient activation resulting in proliferation (*Freed et al., 2017*). Recent cryo-EM studies of EGFR-bound EGF and TGFα revealed that even the high-affinity dimers induce a range of EGFR homodimer conformations differing at receptor intermonomer angles, which might explain distinct functional outcomes downstream from these receptor complexes (*Huang et al., 2021*). These analyses directly link structural differences to functional EGFR outputs, posing a question how of this regulation looks for other ligands and other HER receptor combinations.

Different HER4 ligands were reported to induce unique signaling outputs via activation of HER4 homodimers. Specifically, the two high-affinity HER4 ligands BTC and NRG1β were shown to be different, with BTC more efficiently activating the ERK pathway while NRG1β activated the AKT pathway more potently (*Sweeney et al., 2000*). Our structures of HER4 ectodomain dimers bound to NRG1β and BTC presented here show that in both homodimers there are notable scissor-like movements around the dimerization arms with different intermonomer angles between the two ligands. It is possible that these different dimer conformations influence the stability and consequently signaling outputs emanating from these HER4 homodimers. While these structural differences are seemingly small, they are reminiscent of the ones observed in EGFR homodimers, bound to its two high-affinity ligands, EGF and TGFα (*Huang et al., 2021*).

In contrast to the effects that BTC or NRG1β have on stabilizing more diverse conformational ensembles of the HER4 homodimers, their complexes with the HER2/HER4 heterodimers showed no discernable structural differences. Strikingly in the HER2/EGFR heterodimer, an even more diverse set of growth factors: high-affinity EGF and low-affinity EREG, also failed to stabilize different dimer conformations (*Bai et al., 2023*). Altogether, these structural analyses show that EGFR and HER4 receptors sample a wider selection of active homodimers states that can be exploited by different ligands and might be better conduits for ligand-specific signaling responses that their respective HER2 heterodimers.

## Receptor glycosylation

N-linked glycosylation of receptor tyrosine kinases plays a crucial role in their maturation, stability, and regulation of their interaction with ligands, the extracellular matrix and other membrane proteins (*Duarte et al., 2022*; *Contessa et al., 2008*). HER receptors are heavily glycosylated and their aberrant glycosylation patterns have been associated with diseases such as cancer and promoting drug resistance (*Zhen et al., 2003*; *Rodrigues et al., 2021*; *Britain et al., 2018*; *Peiris et al., 2017*).

Glycosylation patterns on HER receptors can modulate their dimerization propensity. For example, a mutation of the N418 glycosylation site on HER3 promotes its ligand-independent association with HER2 (*Yokoe et al., 2007*). Likewise, mutation of N579 on EGFR drives its ligand-independent activation as well as increases its affinity for ligands (*Whitson et al., 2005*). However, the molecular mechanisms behind most of these effects are poorly understood, mostly because the majority of HER ectodomain structures have been solved by X-ray crystallography using heavily deglycosylated receptor fragments (*Garrett et al., 2002*; *Liu et al., 2012*; *Ogiso et al., 2002*). Recently published cryo-EM analyses of HER receptor samples purified with intact glycosylation, also did not reveal insights into glycan-mediated interactions, perhaps due to their flexible and/or heterogeneous nature in these complexes (*Bai et al., 2023*; *Huang et al., 2021*).

To our knowledge, the cryo-EM maps of HER4 homodimers and HER2/HER4 heterodimers offer the first glimpse into extensive glycan-mediated contacts in an active HER receptor dimer. The observed interactions might explain some of the stabilizing effects of HER receptor glycosylation previously suggested (*Motamedi et al., 2022*). We identified glycans in both HER2 and HER4 that directly connect their domains I and II and the HER4-specific interdomain glycan connections between domains II and IV. Such glycosylation modifications would be expected to stabilize the extended conformations of HER2 and HER4 receptors, although their effect on tethered states cannot be excluded. Most remarkably, our structures of HER4 homodimer ectodomains reveal reasonably well-resolved glycans between N548 of one receptor monomer and N358 of the other, pointing to the potential importance of these interactions in stabilizing the homodimer. In contrast, we have not observed a direct inter-receptor connection for HER2/HER4 heterodimers due to the absence of respective glycosylation sites in HER2. It is tempting to speculate that the particularly strong propensity for HER4 homodimerization over heterodimerization with HER2 that we see in our reconstitution experiments is, at least partially, rooted in the missing glycan-mediated stabilization of the heterodimer.

In recent years several studies of the effects of HER receptor glycosylation on their structure and signaling have been conducted using molecular dynamics (MD) generating models on how glycans contribute to receptor stability and its interactions with the membrane (*Taylor et al., 2017*; *Arkhipov et al., 2013*; *Kaszuba et al., 2015*; *Azimzadeh Irani et al., 2017*). Most recently, MD simulations conducted on the HER4/EGFR heterodimer models have suggested that the glycans present on HER4 N358 and N548, as well as EGFR N361 (which is equivalent to HER4 N358), form a connection in the dimerization interface that effectively stabilizes the heterodimer (*Motamedi et al., 2022*). Our current study provides the first direct experimental evidence that these glycan interactions are operative at the level of HER4 homodimers. Moreover, we analyzed the published EGFR cryo-EM maps (*Huang et al., 2021*) and noticed that the inter-receptor glycans also appear in EGFR homodimers between N361 and the more membrane-proximal N603. Altogether, our analysis points to an important role, and potential conserved mechanisms by which glycosylation contributes to the HER dimer interfaces (*Figure 4—figure supplement 1c–d*).

In summary, our structural analysis provides new knowledge on HER receptor activation by their different growth factor ligands and mechanistic distinctions of their homodimeric versus heterodimeric pairings. Through this, our findings reveal a greater aptitude of HER homodimers to differentiate between biased agonists, at least as compared to HER2-containing heterodimers. Our structures for the first time reveal extensive intra and interdomain glycan contacts at the active HER dimer interface and have the potential to further understanding of how glycosylation can be leveraged for the design of better HER-targeted therapeutics, and how it can contribute to drug resistance.

## Materials and methods
### NRG1β and BTC expression and purification

NRG1β and BTC were expressed and purified as described previously for NRG1β (*Liu et al., 2012*; *Diwanji et al., 2021*; *Trenker et al., 2022*). An HRV-3C cleavable Thyrodoxin A (TrxA) was fused to the EGF-like domain of NRG1β (residues 177–236, NRG1 isoform 6, UniProt: Q02297-6; numbering includes the signal peptide) or BTC (residues 64–117, UniProt: P35070; numbering includes the signal peptide) with C-terminal Flag and 6x-Histidine tags and subsequently cloned into a p32A vector (Millipore Sigma). The TrxA-3C-ligand-Flag-6xHis construct was transformed into *E. coli* Origami B (DE3) pLysS (Millipore Sigma 70839), grown at 37 °C in Terrific Broth until an OD of ~1.0–1.5, and induced

with 1 mM Isopropyl b-d-1-thiogalactopyranoside (IPTG, Goldbio) overnight at room temperature. Cells were harvested the next day, pelleted, flash-frozen, and stored until purification. For purification, cells were resuspended in ligand lysis buffer (50 mM Tris-HCl pH 7.4, 150 mM NaCl, 20 mM imidazole, 1 mM phenylmethylsulfonyl fluoride (PMSF), and protease inhibitors (cOmplete, Roche)) and sonicated until thoroughly lysed. Lysate was then clarified by ultracentrifugation, syringe filtered through 0.44 μm filters, and incubated with Ni-NTA resin (Thermo Fisher Scientific) overnight at 4 °C. The Ni-NTA resin was washed by gravity through 20 column volumes (CVs) of ligand wash buffer (50 mM Tris-HCl pH 7.4, 150 mM NaCl) containing 20 mM imidazole, then 10 CVs of ligand wash buffer containing 50 mM imidazole, and finally eluted with three CVs of ligand wash buffer containing 300 mM imidazole. Imidazole in the eluate was reduced <30 mM over a 10 K MWCO concentrator and subsequent dilution with ligand wash buffer. The eluted protein was cleaved overnight with 3 C protease at 4 °C. To remove cleaved TrxA, the elution was again applied to equilibrated Ni-NTA resin, incubated, washed, and eluted as described above. The elution containing NRG1β was concentrated with a 3 K cutoff and applied on an S200 10/300 increase column (GE Healthcare). Protein content of the major peak was stored in aliquots at –80 °C for subsequent receptor purifications.

## Receptor expression

Human HER2 was expressed as previously described (*Diwanji et al., 2021*). HER2 with a C-terminal tail truncation (Δ1030–1255) followed by maltose binding protein (MBP) and twin-strep tags was cloned into pFastBac1 with a CMV promoter (Thermo Fisher Scientific). Point mutations were introduced in the HER2 kinase domain, G778D, and V956R, to confer Hsp90 independence for improved yields and to position the HER2 kinase domain in the receiver position of an asymmetric HER kinase dimer, respectively. For heterodimer formation, human HER4 JM-A CYT-1 isoform with a C-terminal tail truncation (D1029 – 1308) followed by a twin-strep tag was cloned in pFastBac with a CMV promoter. A I712Q mutation was introduced to position HER4 in the activator position in a HER2/HER4 heterodimer (mutagenesis primer listed in the 'cell-based assays' section). The HER2 and HER4 constructs were each transfected into 30 ml or 60 ml of Expi293F mammalian suspension cells (Thermo Fisher Scientific, A14527,RRID:CVCL_D615) cultured to $4 \times 10^6$ cells/ml at 37 °C, 8% $CO_2$ following the standard expression protocol with 1 μg DNA/ml cultures. 10 mM canertinib (MedChemExpress) in DMSO was added to HER2 cultures 16–18 hr post-transfection to a final concentration of 10 μM along with ExpiFectamine 293 Transfection Kit enhancers 1 and 2. Cells were harvested, flash frozen, and stored at –80 °C 24 hr after the addition of enhancers. For homodimer formation, full-length, untagged wild-type HER4 JM-A CYT-1 isoform was cloned into a pCDNA4TO (Thermo Fisher Scientific) and transfected into Expi293F cultures as described above. If utilized, a final concentration of 10 μM afatinib (MedChemExpress) was added as described above for canertinib.

## HER2/HER4 heterodimer and HER4 homodimer purification

For heterodimer purification, cell pellets from 120 ml suspension cultures for each receptor were resuspended with the lysis buffer (50 mM Tris-HCl pH 7.4, 150 mM NaCl, 1 mM NaVO₃, 1 mM NaF, 1 mM EDTA, protease inhibitors (cOomplete, Roche), DNAse I (Roche), and 1% DDM (Inalco)) and lysed for 2 hr by gentle rocking at 4 °C. Lysate was clarified by centrifugation at 4000 g for 10 min at 4 °C. Purified EGF-like domain of NRG1β or BTC was incubated with anti-DYKDDDDK G1 affinity resin (Genscript, short anti-Flag) for 1 hr at 4 °C and serially washed 3 x with Buffer A (50 mM Tris-HCl pH 7.4, 150 mM NaCl). Clarified HER2 and HER4 receptor lysates were mixed and incubated O/N in batch mode at 4 °C with ligand-coated Flag beads. Ligand-coated anti-Flag beads were serially 3 x washed with Buffer A containing 0.5 mM DDM (Anatrace) and eluted with Buffer A containing 0.5 mM DDM and 250 μg/ml of Flag peptide (SinoBiological). The eluate was then applied to amylose resin in batch mode for 2 hr, washed serially 3 x with Buffer B (50 mM HEPES pH 7.4, 150 mM NaCl) containing 0.5 mM DDM and eluted with amylose elution buffer (Buffer B containing 0.5 mM DDM and 20 mM maltose) O/N at 4 °C. The eluate was concentrated to 0.4 ml with a 100 kDa concentrator (Amicon), mildly crosslinked in 0.2% glutaraldehyde (Electron Microscopy Sciences) for 40 min on ice and quenched by addition of 40 μl of 1 M Tris pH 7.4. The sample was loaded on a Superose6 increase 10/300 (GE Healthcare) gel filtration column pre-equilibrated with Buffer A containing 0.5 mM DDM and 0.5 ml fractions were collected. Peak fractions corresponding to the heterodimer sample were pooled, and concentrated to ~0.1 μM with a 100 kDa concentrator for EM grid preparation. For the

purification of liganded HER4 homodimers, the same purification protocol for the ligand-mediated receptor pulldown was followed. After elution from anti-Flag resin, the receptor was concentrated, crosslinked with glutaraldehyde, and subjected to gel filtration as described above. Peak fractions corresponding to the homodimer sample were pooled and concentrated with a 100 kDa concentrator to 0.1 µM for EM grid preparation or flash frozen in liquid nitrogen and stored at –80 °C.

## Electron microscopy sample preparation and imaging

For negative stain EM, fractions corresponding to heterodimer were applied to negatively glow-discharged carbon-coated copper grids, stained with 0.75% uranyl-formate, and imaged on an FEI-Tecnai T12 with an 4 k CCD camera (Gatan). The resulting negative stain micrographs were assessed for particle homogeneity and particle density. This analysis was used to determine the target concentration for cryo-EM with graphene oxide grids which typically required 2–5 x negative stain concentrations.

For cryo-EM, 3 µl of purified and concentrated heterodimer sample (as empirically determined by negative stain, typically around ~0.1 µM) was applied to graphene-oxide coated Quantifoil R1.2/1.3 300 mesh Au holey-carbon grids prepared as previously described (*Diwanji et al., 2021*), blotted using a Vitrobot Mark IV (FEI) and plunge frozen in liquid ethane (no glow discharge, 30 s wait time, room temperature, 100% humidity, 5–7 s blot time, 0 blot force).

Grids were imaged on a 300-keV Titan Krios (FEI) with a K3 direct electron detector (Gatan) and a BioQuantum energy filter (Gatan) operating with an energy slit width of 20 eV. Data for HER2/HER4/NRG1β, HER2/HER4/BTC, and HER4/BTC were collected in super-resolution mode at a physical pixel size of 0.835 Å/pix with a dose rate of 16.0 e$^-$ per pixel per second (operated in CDS mode) and a total dose of 45.8 e$^-$/Å$^2$. Images were recorded with a 2.0 s exposure over 80 frames with a dose of 0.57 e$^-$/Å$^2$/frame at 0.025 s/frame. Data for HER4/NRG1β were collected in super-resolution mode at a physical pixel size of 0.835 Å/pix with a dose rate of 16.0 e$^-$ per pixel per second (operated in CDS mode) and a total dose of 68.7 e$^-$/Å$^2$. Images were recorded with a 3.0 s exposure over 120 frames with a dose of 0.57 e$^-$/Å$^2$/frame at 0.025 s/frame.

## Image processing and 3D reconstruction

Raw movies were corrected for motion and radiation damage with MotionCor2 (*Zheng et al., 2017*) and the resulting sums were imported in CryoSPARC v2 (HER2/HER4/BTC, HER4/NRG1β, HER4/BTC), or CryoSPARC v4 (HER2/HER4/NRG1β) (*Punjani et al., 2017*). Micrograph CTF parameters were estimated with the patch CTF estimation job in CryoSPARC v2 (HER2/HER4/BTC, HER4/NRG1β, HER4/BTC), or CryoSPARC v4 (HER2/HER4/NRG1β). Particles were picked using a template picker with low-pass filtered (20–25 Å) 2D templates created from imported HER receptor dimer volumes, initially from published HER2/HER3/NRG1β heterodimers. HER4 homodimer particles were template-picked a second time using 2D class averages as a template obtained from a first round of picking and processing (see processing flow charts for sample-specific details). The resulting picks were extracted with a box size of 384 pix (320.64 Å) with 2 x Fourier cropping and subjected to initial 2D classification to remove obviously poor classes and picks containing lines from visible graphene-oxide flakes. More than 90% of picks were selected and subjected to *ab initio* reconstruction into three classes. In all datasets, this resulted in 'junk' classes without recognizable HER receptor features. To purify this particle set, all 2D-selected particles were subjected to two rounds of heterogeneous refinement containing a HER receptor dimer volume (imported from previous datasets or obtained from this dataset in a previous round of processing using the same overall workflow) and 3 'junk' classes. Particles sorted into the HER receptor dimer volume were subjected to *ab initio* reconstruction into one or two classes, depending on which resulted in better resolution downstream, followed by heterogeneous refinement (in two classes) and non-uniform refinement (see processing flow charts for sample-specific details). Once reasonable reconstructions were obtained (as judged by the FSC (Fourier Shell Correlation) curve shape), unbinned particles were re-extracted and subjected to *ab initio* reconstruction, heterogeneous refinement or 2D classification/selection, and finally non-uniform refinement to achieve reconstructions with the highest resolution. The map of HER2/HER4/NRG1β was manually sharpened with an applied B-factor of –85. The final reconstructions of HER2/HER4/NRG1β and HER2/HER4/BTC used for model building included 289,192 and 148,541 particles and resulted in an overall resolution of 3.31 Å and 4.27 Å by Gold Standard-Fourier Shell Correlation

(GS-FSC) cutoff of 0.143, respectively. The final reconstructions of HER4/NRG1β and HER4/BTC used for model building included 205,726 and 274,540 particles and attained a GS-FSC resolution of 3.38 and 3.70 Å with C1 symmetry, respectively. Applying C2 symmetry in the final non-uniform refinement run improved the resolution to 3.26 and 3.49 Å. However, due to imperfect C2 symmetry observed in our C1 reconstructions, most of the reported analysis was performed using C1 reconstructions. Each map was assessed for local and directional resolutions in cryoSPARC v4 and 3DFSC (*Tan et al., 2017*) serves, respectively. Except for the HER2/HER4/BTC, extracellular domains I-III achieved the highest local resolutions (~3 Å) while that of domain IV varied from 4 to above 7 Å suggesting that a high degree of flexibility exists closer to the transmembrane domains. Unless specifically mentioned here or in the processing workflow, default parameters in CryoSPARC were used at each processing step.

## Model refinement and validation

For HER2/HER4, an initial model was generated by placing the HER2/HER3/NRG1β heterodimer (PDB ID: 7MN5) into the HER2/HER4/NRG1β map and replacing HER3 with a model of HER4 (PDB ID: 3U7U, HER4 chain C and NRG1β chain I) after alignment onto HER3 using UCSF ChimeraX. The model was fit into the density with a FastRelax Rosetta protocol in torsion space, refined once with PHENIX (*Adams et al., 2010*) real-space refinement, and further modeled using iterative rounds of ISOLDE (*Croll, 2018*) and the FastRelax Rosetta protocol in torsion space (*Maguire et al., 2021*; *Fleishman et al., 2011*; *Khatib et al., 2011*). Per atom B-factors were assigned in Rosetta indicating the local quality of the map around that atom. Glycans were built into the density onto a well-refined model using the Carbohydrate module in Coot (*Emsley et al., 2010*) for mammalian proteins and refined with the Rosetta glycan refinement protocol (*Frenz et al., 2019*). After glycan addition, the model was once more refined in ISOLDE, and the Rosetta FastRelax protocol in torsion space, and main and side chains for domains I-III were inspected for final corrections in Coot. Model statistics were routinely assessed in PHENIX (*Adams et al., 2010*) and glycan geometries were cross-validated in Privateer (*Agirre et al., 2015*).

For the HER2/HER4/BTC heterodimer, the final model of HER2/HER4/NRG1β was placed into the cryo-EM density and NRG1β was replaced with a model of the BTC EGF-like domain originally obtained from the AlphaFoldDB (AF-P35070-F1) by alignment in UCSF ChimeraX. The model was fit into the density with a FastRelax Rosetta protocol in torsion space, further refined in ISOLDE and main and side chains for domains I-III were inspected for final corrections in Coot. Per atom, B-factors were assigned in Rosetta indicating the local quality of the map around that atom.

For the HER4/NRG1β homodimer, an initial model was created by placing HER4/NRG1β models from a crystal structure (PDB ID: 3U7U, HER4 chains C+D, NRG1β chains I+K) into the cryo-EM density and running the FastRelax Rosetta protocol in torsion space. The model was further refined using iterative rounds of ISOLDE and the FastRelax Rosetta protocol in torsion space. Per atom B-factors were assigned in Rosetta indicating the local quality of the map around that atom. Glycans were built into the density onto a well-refined model using the Carbohydrate module in Coot (*Emsley et al., 2010*) for mammalian proteins and refined with the Rosetta glycan refinement protocol (*Frenz et al., 2019*). After glycan addition, the model was once more refined in ISOLDE, and the Rosetta FastRelax protocol in torsion space and main and side chains for domains I-III were inspected for final corrections in Coot. Model statistics were routinely assessed in PHENIX (*Adams et al., 2010*) and glycan geometries were cross-validated in Privateer (*Agirre et al., 2015*).

A HER4/BTC homodimer model was created by placing the final model of HER4/NRG1β into the cryo-EM density and NRG1β was replaced with a model of the BTC EGF-like domain obtained from the AlphaFoldDB (AF-P35070-F1) by alignment in UCSF ChimeraX. The model was fit into the density with a FastRelax Rosetta protocol in torsion space, further refined in iterative rounds of ISOLDE the Rosetta FastRelax protocol in torsion space, and main and side chains for domains I-III were inspected for final corrections in Coot. Per atom B-factors were assigned in Rosetta indicating the local quality of the map around that atom. Model statistics were routinely assessed in PHENIX (*Adams et al., 2010*) and glycan geometries were cross-validated in Privateer (*Agirre et al., 2015*).

## 3D classification analysis

3D classification analysis was performed using the heterogeneous refinement function in cryoSPARC v4. For each hetero- or homodimer, the particle stacks from their final reconstructions were used as

input for heterogeneous refinement together with four identical respective volumes as initial models. The same respective atomic model was fit into the four resulting volumes from the classification using the Rosetta FastRelax protocol in torsion space resulting in four different models for each 3D classification. Models were aligned on the same chain for visualization and intermonomer angles for each torsion-relaxed model were measured as described below. To ensure robustness, the analysis was repeated using the same particle stacks for HER4/NRG1β and HER4/BTC homodimers but using starting volumes of the other homodimer (HER4/NRG1β particles, HER4/BTC starting volumes and vice versa). This yielded similar results indicating that 3D classes are being determined by the particles, not the starting volumes.

## Structure analysis

UCSF ChimeraX was used to determine the interface residues, H-bonds, and interface area between two chains of a model. The command to measure the buried area between model 1 chain A and chain B is: (*measure buried area #1 /A with Atoms2 #1/B*). This interface area was then multiplied by 2 to obtain the total buried surface area (BSA) of both proteins. Prior to the measurements, hydrogens were added to all models (*addh*). BSA for HER receptor dimerization interfaces was done for residues 1–450. Polypeptide backbone overlays and determinations of RMSDs between two chains/models was performed using UCSF ChimeraX using the matchmaker command. RMSD values reported are across all pairs of the sequence alignment. Intermonomer angles were determined using UCSF ChimeraX by defining an axis through monomer 1 A and 1B and measuring the angle between the two axes (Commands: *define axis #1 /A, define axis #1/B, angle #1.2 #1.3*). Glycans were removed from the polypeptide chain for this analysis.

## Cell-based assays

Untagged full-length human HER4 was cloned into a pcDNA4TO expression vector. Full-length human HER2 tagged with the C-terminal 3xFLAG tag in a pcDNA4TO expression vector was kindly provided by Mark Moasser. HER2 I714Q, HER2 V956R, HER4 I712Q, and HER4 V954R were introduced into pcDNA vectors by site-directed mutagenesis. HER2 constructs in cell-based activity assays do not feature the G778D mutation. Untagged HER2 and HER4 constructs, and their mutants, were further cloned into a pMSCV retroviral vector, kindly provided by James Fraser, using standard PCR methods and Gibson assembly. GS-arm mutations replacing HER2 (residues: A[270]LVTYNTDTFESMPNP[285]) and HER4 (residues: Q[264]TFVYNPTTFQLEHNF[279]) with an alternating sequence of glycine and serine residues: 'GSGSGSGSGSGSGSGSGS' were introduced into pMSCV vectors via PCR and Gibson assembly.

Mutagenesis primer sequences to introduce point mutations and GS-arm mutations are listed below (5' ->3'):

> HER4-GS for:
> GTGTTACTCAGTGTCCCGGCTCTGGCTCTGGGTCGGGCTCTGGGTCGGGCTCTGGGT
> CTGGGTCGAATGCAAAGTACACATATGGAG
> HER4-GS rev:
> CTCCATATGTGTACTTTGCATTCGACCCAGACCCAGAGCCCGACCCAGAGCCCGACCCAG
> AGCCAGAGCCGGGACACTGAGTAACAC
> HER2-GS for:
> GAGCTGCACTGCCCAGGCTCTGGCTCTGGGTCGGGCTCTGGGTCGGGCTCTGGGTC
> TGGGTCGGAGGGCCGGTATACATTCGGC
> HER2-GS rev:
> GCCGAATGTATACCGGCCCTCCGACCCAGACCCAGAGCCCGACCCAGAGCCCGACC
> CAGAGCCAGAGCCTGGGCAGTGCAGCTC
> HER2 V956R for:
> GATGTCTACATGATCATGAGGAAATGTTGGATGATTGAC
> HER2 V956R rev:
> GTCAATCATCCAACATTTCCTCATGATCATGTAGACATC
> HER2 I714Q for:
> CAACCAGGCGCAGATGCGGCAGCTGAAAGAGACGGAGCTG
> HER2 I714Q rev:
> CAGCTCCGTCTCTTTCAGCTGCCGCATCTGCGCCTGGTTG

HER4 V954R for:
CTATTGACGTTTACATGGTCATGCGCAAATGTTGGATGATTGATGCTG
HER4 V954R rev:
CAGCATCAATCATCCAACATTTGCGCATGACCATGTAAACGTCAATAG
HER4 I712Q for:
CACCCAATCAAGCTCAACTTCGTCAGTTGAAAGAAACTGAGCTGAAGAG
HER4 I712Q rev:
CTCTTCAGCTCAGTTTCTTTCAACTGACGAAGTTGAGCTTGATTGGGTG

For COS7 (CRL-1651, RRID:CVCL_0224) transient transfection experiments, $0.12 \times 10^6$ COS-7 cells were seeded into each well of a six-well plate and transfected with 1 µg total DNA using Lipofectamine p3000 (ThermoFisher Scientific). Cells were rinsed in PBS 5 hr post-transfection, serum-starved for 16 hr and, if applicable, stimulated with 10 nM NRG1β(PeproTech) or BTC (PeproTech) for 10 min at 37 °C. Cells were then washed with ice-cold PBS two times and lysed in 300 µl RIPA buffer (50 mM TRIS pH 8, 150 mM NaCl, 1% NP40, 0.5% sodium deoxycholate, 0.1% SDS, 1 mM EDTA, Roche Complete protease inhibitors, DNAse, 1 mM sodium orthovanadate, 1 mM sodium fluoride) on ice for 30 min. Lysates were transferred into 1.5 ml microcentrifuge tubes, spun at $15,000 \times g$ for 3 min and supernatants were transferred into fresh tubes and mixed with the SDS loading dye. HER2, HER4, phospho-HER2 (pY1221/1222) and pHER4 (1284) levels were determined by Western blot using following antibodies: rabbit anti-HER4 (Cell Signaling, Cat# 111B2, 1:1000, RRID:AB_2099883), rabbit anti-phospho-Y1283 HER4 (Cell Signaling, Cat# 21A9, 1:1000, RRID:AB_2099987), rabbit anti-HER2 (Cell Signaling, Cat# D8F12, 1:1000, RRID:AB_10557104), rabbit anti-phospho-Y1221/1222 HER2 (Cell Signaling, Cat# 2249, 1:1000, RRID:AB_2099241), anti-rabbit IgG HRP-linked antibody (Cell Signaling, Cat# 7074, 1:5000, RRID:AB_2099233).

GS-arm experiments were performed in NR6 cells kindly provided by Mark Moasser that were transduced with retroviral vectors produced in PlatE packaging cells (Cell Biolabs, RV-101, RRID:CVCL_B488). NR6 cells (RRID:CVCL_6694) were maintained in DMEM/F12 (1:1) with 2 mM glutamine, PenStrep, and 10% FCS. Packaging PlatE cells were maintained in DMEM, PenStrep, 10% FCS, 10 µg/ml blasticidin and 1 µg/ml puromycin. To produce retrovirus, $1 \times 10^6$ PlatE cells were plated in a 60 mm dish. Medium was replaced with NR6 media the next day and cells were transfected with 3 µg pMSCV DNA constructs using Lipofectamine P3000. Retroviral supernatants were harvested 48 hr post-transfection, filtered through a syringe filter unit with 0.45 µm filter size, and added to NR6 cells seeded at $0.1 \times 10^6$ cells/well in a 12 well plate the prior day. For spin infection, 8 µg/ml polybrene (EMD Millipore) was added and cells were spun for 90 min at $800 \times g$ at room temperature. Cells were then incubated overnight and the infection medium was replaced with NR6 medium containing 2 µg/ml puromycin for selection for one week.

For NR6 signaling assays, respective stable cell lines were plated in six-well plates at a density of $\sim 1 \times 10^5$ cells/well (70–80% confluency). The next day, cells were serum-starved for 4 hr and, if applicable, stimulated with 10 nM NRG1β. Cells were then washed with ice-cold PBS two times and lysed in 300 µl RIPA buffer (50 mM TRIS pH 8, 150 mM NaCl, 1% NP40, 0.5% sodium deoxycholate, 0.1% SDS, 1 mM EDTA, Roche Complete protease inhibitors, DNAse, 1 mM sodium orthovanadate, 1 mM sodium fluoride) on ice for 30 min. Lysates were transferred into 1.5 ml microcentrifuge tubes, spun at $15,000 \times g$ for 3 min and supernatants were transferred into fresh tubes and mixed with SDS loading dye for Western blot analysis. Membranes were cut around the 70 kDa marker band and HER2, HER4, phospho-HER2 (pY1221/1222) and pHER4 (1284) levels in lysates were by antibody staining as described above. The membrane containing lower molecular weight proteins were stained for Erk, pErk, AKT, pAKT, and Actin using the following antibodies: mouse anti-Erk (Cell Signaling, L34F12, 1:1000, RRID:AB_390780) rabbit anti-phospho-p44/42 MAPK (Erk1/2) (Cell Signaling, 9101, 1:1000, RRID:AB_331646), mouse anti-AKT (Cell Signaling, 40D4, 1:1000, RRID:AB_2273787), rabbit anti-phospho-AKT recognizing phosphorylated serine S473 (Cell Signaling, 1:1000, #9271, RRID:AB_329825), mouse anti-β-Actin (Santa Cruz Biotechnology, sc047778, 1:1000), anti-rabbit IgG HRP-linked antibody (Cell Signaling, 1:5000, #7074, RRID:AB_2099233), anti-mouse IgG HRP-linked (ECL, NXA931V, 1:5000).

## Acknowledgements

We thank members of the Verba and Jura labs for their helpful discussions, and E Linossi for critical comments on the manuscript. We thank D Bulkley, G Gilbert, and E Tse from the UCSF EM facility for their assistance with data collection. We thank M Moasser for kindly providing NR6 cells and J Fraser and G Estevam for providing the pMSCV constructs. This work was funded through UCSF Program for Breakthrough Biomedical Research to KV and NJ, NIH/NIGMS R35-GM139636 to NJ, NIH/NCI U54CA274502 to NJ, DFG German Research Foundation GZ: TR 1668/1-1 to RT and NIH/NCI 1F30CA247147 to DD.

## Additional information

### Competing interests

Natalia Jura: Reviewing editor, *eLife*. The other authors declare that no competing interests exist.

### Funding

| Funder | Grant reference number | Author |
| --- | --- | --- |
| National Institutes of Health | R35GM139636 | Natalia Jura |
| National Cancer Institute | U54AI170792 | Natalia Jura |
| Deutsche Forschungsgemeinschaft | GZ: TR 1668/1-1 | Raphael Trenker |
| National Institutes of Health | 1F30CA247147 | Devan Diwanji |

The funders had no role in study design, data collection and interpretation, or the decision to submit the work for publication.

### Author contributions

Raphael Trenker, Conceptualization, Data curation, Formal analysis, Validation, Investigation, Visualization, Methodology, Writing - original draft, Writing - review and editing; Devan Diwanji, Conceptualization, Investigation, Methodology, Writing - review and editing; Tanner Bingham, Methodology; Kliment A Verba, Natalia Jura, Conceptualization, Resources, Data curation, Formal analysis, Supervision, Funding acquisition, Investigation, Methodology, Writing - original draft, Project administration, Writing - review and editing

### Author ORCIDs

Raphael Trenker ⬤ http://orcid.org/0000-0003-1748-0517
Devan Diwanji ⬤ http://orcid.org/0000-0002-4285-435X
Kliment A Verba ⬤ http://orcid.org/0000-0002-2238-8590
Natalia Jura ⬤ http://orcid.org/0000-0001-5129-641X

Reviewer 2 (Public Review): https://doi.org/10.7554/eLife.92873.3.sa1
Reviewer #1 (Public Review): https://doi.org/10.7554/eLife.92873.3.sa2
Author Response https://doi.org/10.7554/eLife.92873.3.sa3

## Additional files

### Supplementary files

• MDAR checklist

### Data availability

All structures were deposited in the Protein Data Bank (PDB) and Electron Microscopy Data Bank (EMDB) with the following identifiers: PDB: 8U4L and EMD-41886 for HER2/HER4/NRG1, EMD-41885

and PDB: 8U4K for HER2/HER4/BTC, EMD-41883 and PDB:8U4I for HER4/NRG and EMD-41884 and PDB:8U4J for HER4/BTC.

The following datasets were generated:

| Author(s) | Year | Dataset title | Dataset URL | Database and Identifier |
|-----------|------|---------------|-------------|-------------------------|
| Trenker R, Diwanji D, Verba K, Jura N | 2024 | HER2/HER4 heterodimer bound to NRG1 | https://www.rcsb.org/structure/8U4L | RCSB Protein Data Bank, 8U4L |
| Trenker R, Diwanji D, Verba K, Jura N | 2024 | HER2/HER4 heterodimer bound to NRG1 | https://www.ebi.ac.uk/emdb/EMD-41886 | Electron Microscopy Data Bank, EMD-41886 |
| Trenker R, Diwanji D, Verba K, Jura N | 2024 | HER2/HER4 heterodimer bound to BTC | https://www.rcsb.org/structure/8U4K | RCSB Protein Data Bank, 8U4K |
| Trenker R, Diwanji D, Verba K, Jura N | 2024 | HER2/HER4 heterodimer bound to BTC | https://www.ebi.ac.uk/emdb/EMD-41885 | Electron Microscopy Data Bank, EMD-41885 |
| Trenker R, Diwanji D, Verba K, Jura N | 2024 | HER4 homodimer bound to NRG1 | https://www.rcsb.org/structure/8U4I | RCSB Protein Data Bank, 8U4I |
| Trenker R, Diwanji D, Verba K, Jura N | 2024 | HER4 homodimer bound to NRG1 | https://www.ebi.ac.uk/emdb/EMD-41883 | Electron Microscopy Data Bank, EMD-41883 |
| Trenker R, Diwanji D, Verba K, Jura N | 2024 | HER4 homodimer bound to BTC | https://www.rcsb.org/structure/8U4J | RCSB Protein Data Bank, 8U4J |
| Trenker R, Diwanji D, Verba K, Jura N | 2024 | HER4 homodimer bound to BTC | https://www.ebi.ac.uk/emdb/EMD-41884 | Electron Microscopy Data Bank, EMD-41884 |

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
