## [Editor Report · eLife assessment]

This manuscript describes structures of HER4 homo- and HER4/HER2 hetero-dimer complexes using single particle cryo-EM. This **important** work **convincingly** describes new structural details of these complexes that expand our understanding of their function. This work will be of interest to researchers working on cell surface signalling and kinase activity.

---

## [Referee Report · Reviewer #1 (Public Review)]

Trenker et al. report cryo-EM structures of HER4/HER2 heterodimers and HER4 homodimers bound to Neuregulin-1β (Nrg1β) and Betacellulin (BTC). As observed for prior cryo-EM structures of full-length or near full-length HER-family receptors only the extracellular regions are visualized, presumably owing to flexibility in the relative orientation of extra- and intra-cellular regions. The authors observe no appreciable differences between Nrg1β and BTC bound heterodimers, both ligands in this case being high-affinity ligands, and modest "scissor-like" differences in the subunit relationships in HER4 homodimers with Nrg1β and BTC bound.

The authors also show that, as they showed for HER3, the HER4 dimerization arm is not indispensable for forming heterodimers with HER2 despite the HER4 dimerization arm forming a more canonical interaction with HER2. Perhaps most interestingly, the authors observe glycan interactions that appear to stabilize intra- and inter-subunit interactions in HER4 homodimers but that inter-subunit glycans are not present in HER2/HER4 heterodimers. The authors speculate that these glycan interactions may contribute to the apparent propensity of HER4 to homodimerize vs. heterodimerize with HER2.

---

## [Referee Report · Reviewer #2 (Public Review)]

With the data presented in this manuscript, the authors help complete the set of high resolution HER2- associated complex heterodimer structures as well as HER4 homodimer structures in the presence of NRG1b and BTC. Purification of HER2-HER4 heterodimers appears to be inherently challenging due to the propensity of HER4 to form homodimers. The authors have used an effective scheme to isolate these HER2-HER4 heterodimers and have employed graphene-oxide grid chemistry to presumably overcome the issues of low sample yield for solving cryo-EM structures of these complexes. The authors conclude HER2-HER4 heterodimers with either ligand is conformationally homogeneous relative to the HER4 homodimers. The HER2-HER4 heterodimers also appear to be better stabilized compared to other published HER2 heterodimers. The ability to model glycans in the context of HER4 homodimers is exciting to see and provides a strong rationale for the stability of these structures. Overall, the work is of great interest and the methods described in this work would benefit a wide variety of structural biology projects.

---

## [Author Response]

The following is the authors’ response to the original reviews.

**Reviewer #1 (Public Review):**
Trenker et al. report cryo-EM structures of HER4/HER2 heterodimers and HER4 homodimers bound to Neuregulin-1b (Nrg1b) and Betacellulin (BTC). As observed for prior cryo-EM structures of full-length or near full-length HER-family receptors only the extracellular regions are visualized, presumably owing to flexibility in the relative orientation of extra- and intra-cellular regions. The authors observe no appreciable differences between Nrg1b and BTC bound heterodimers, both ligands, in this case being high-affinity ligands, and modest "scissor-like" differences in the subunit relationships in HER4 homodimers with Nrg1b and BTC bound.The authors also show that, as they showed for HER3, the HER4 dimerization arm is not indispensable for forming heterodimers with HER2 despite the HER4 dimerization arm forming a more canonical interaction with HER2. Perhaps most interestingly, the authors observe glycan interactions that appear to stabilize intra- and inter-subunit interactions in HER4 homodimers but that inter-subunit glycans are not present in HER2/HER4 heterodimers. The authors speculate that these glycan interactions may contribute to the apparent propensity of HER4 to homodimerize vs. heterodimerize with HER2.I realize that an important role of reviewers is to provide authors with informed and critical comments, but I found this manuscript a well-written, thoughtful, and important contribution. My only note is that I am not an electron microscopist so have assumed the microscopy has been carried out expertly and rely on other reviewers to vet structure determinations.

We thank the reviewer for sharing our enthusiasm and the positive assessment of our manuscript. We have carefully reviewed the all microscopy-related concerns while responding to the assessment of reviewer #2.

**Reviewer #2 (Public Review):**
With the data presented in this manuscript, the authors help complete the set of high-resolution HER2-associated complex heterodimer structures as well as HER4 homodimer structures in the presence of NRG1b and BTC. Purification of HER2-HER4 heterodimers appears to be inherently challenging due to the propensity of HER4 to form homodimers. The authors have used an effective scheme to isolate these HER2-HER4 heterodimers and have employed graphene-oxide grid chemistry to presumably overcome the issues of low sample yield for solving cryo-EM structures of these complexes. The authors conclude HER2-HER4 heterodimers with either ligand are conformationally homogeneous relative to the HER4 homodimers. The HER2-HER4 heterodimers also appear to be better stabilized compared to other published HER2 heterodimers. The ability to model glycans in the context of HER4 homodimers is exciting to see and provides a strong rationale for the stability of these structures. Overall, the work is of great interest and the methods described in this work would benefit a wide variety of structural biology projects.

We thank the reviewer for their positive assessment of our manuscript.

Major comments:1. The HER2-HER4 heterodimer with BTC appears to be the lowest resolution of the reported structures. Although the authors claim the overall structure is similar to the HER2-HER4 heterodimer with NRG1b, it is therefore unclear whether the lower resolution of the BTC is due to challenging data collection conditions, sample preparation, or conformational dynamics not discernible due to the lower resolution. The authors should minimally clarify where they see the possible issues arising for the lower resolution as this is a key aspect of the work.

The most likely reason for the lower resolution of the HER2/HER4/BTC reconstruction is not the underlying fundamental biology but a certain degree of preferred orientations in the sample, as can be seen from the directional FSC curves in the supplemental materials (Figure S3). We would like to note that while the overall resolution of the HER2/HER4/BTC reconstruction may be comparatively lower than other reconstructions presented in the manuscript, it remains of sufficiently high quality to substantiate our key claims. Specifically, our analysis indicates a close resemblance between the HER2/HER4/BTC reconstruction and the HER2/HER4/NRG reconstruction. For example, individual beta strands can still be well resolved allowing their accurate placement. There may be differences in features at higher resolution than 4.5Å between these two reconstructions which we cannot observe due to the lower resolution of HER2/HER4/BTC map, but these would amount to side chain motions rather than larger secondary structure movement. In the manuscript, we only draw comparisons between domain movements in different heterodimer structures and do not see any conformational variability in the final reconstructions, nor in their 3D classification analyses. Thus, we do not attribute the lower resolution of HER2/HER4/BTC reconstruction to increased dynamics at resolution scales that are discussed in the manuscript. What is more likely, is that variability in data quality, which we commonly observe between different GO grids, contributes to differences in resolution between different samples and potentially to the different orientation distributions. To comment on these possibilities, we added the following text to the manuscript (italic, underlined):

Page 8 top paragraph:

“Despite the diverse sequences of the NRG1β and BTC ligands, the larger-scale domain conformation of the HER2/HER4 heterodimers stabilized by each ligand is identical with only small differences in the ligand binding pockets (Figure 1d). Due to the lower resolution of the HER2/HER4/BTC complex, we cannot exclude the possibility of differences in side-chain arrangements between the two structures. However, we attribute the lower resolution to variability in data collection on GO grids, which we frequently observe, rather than differences in conformational heterogeneity of HER2/HER4/BTC.”

Page 10, second paragraph:

“Our cryo-EM structures of the full-length HER2/HER4 complexes bound to either NRG1β or BTC, did not reveal discernible differences at the receptor dimerization interface and larger-scale domain arrangements (Figure 1d).”

2. For all maps, authors should display Euler angle plots from their final refinements to assess the degree of preferred orientation. Judging by the sphericity, it appears all the structures, except HER2-HER4-BTC, have well-sampled projection distributions. However, a formal clarification would be useful to the reader.

We thank the reviewer for pointing this out. We regarded the 3DFSC curves included in our original submission as sufficient measure for projection distributions. In the revised manuscript, we now also include Euler angle plots from respective CryoSPARC refinements in the supplemental Figures.

3. The authors should also include map-model FSCs to ascertain the quality of the map with respect to model building, as this is currently missing in the submission.

We included map-model FSCs from Phenix validation runs in our supplemental material.

Minor comments:1. With respect to complex formation, is there a reason why HER2 expression is dramatically lower than HER4?

The expression of HER2 and HER4 in Expi293F cells, and consequently the amount of HER2 and HER4 receptors at the beginning of our first purification step, which is the NRG1b-mediated pulldown of HER4, is not noticeably different. After this initial purification step, a significant portion of HER2 is lost due to the fact that HER2/HER4 complexes constitute only a small fraction of the total HER complexes because HER4 homodimers preferentially tend to form. This is the reason why HER4 levels after the first purification step shown on the gel in Figure S1b are significantly higher than those of HER2. In the revised manuscript, in Figure S1d, we now show that both receptors are expressed at a comparable levels at the beginning of purification. In this experiment, levels of HER2-MBP-TS and HER4-TS purified separately from the equivalent volumes of transfected Exp293F cell culture via their shared TS-tags (MBP=Maltose Binding Protein, TS=Twin-Strep) are evaluated on a Coomassie-stained gel. When equal volumes of these elutions are then mixed and either subjected to HER4-directed pulldown using NRG1b-coated Flag-resin (lane 3, Figure S1d of the revised manuscript) or HER2-MBP-directed pulldown using amylose resin in the presence of NRG1b (lane 4, Figure S1d of revised manuscript), none of these pulldowns reveals substantial HER2/HER4 heterodimerization indicating that HER4 homodimerization is favored.

2. Figures S1e authors should clarify if HER2 substitutions are VR alone or do these include GD substitutions as well. These should be suitably clarified in the main text.

The HER2 constructs used in all cellular assays do not include the G778D mutation. We clarified this in Figure S1e, in the Materials and Methods section and in the main text on page 6.

3. The validation reports for all 4 reported structures suggest the user-provided FSC-derived resolutions are different from those calculated by the deposition server. Are the masks deposited significantly different compared to the ones generated within cryoSPARC?

The user-provided FSC-derived resolutions are different from those calculated by the server because the server only calculates resolution of unmasked curves from half maps while we provide the resolution derived from masked FSCs. These were all calculated using masks generated within the respective refinement job in cryoSPARC. However, we did notice that our author-provided FSC curves were from unmasked maps and we replaced the provided unmasked FSCs with masked FSCs as generated in cryoSPARC. These FSC plots in the validation reports now reflect the author-provided resolution in our validation reports and the plots generated by cryoSPARC shown in Figures S2, S3, S9 and S10.

4. For interpretation regarding activation through phosphorylation in Figure 2e, have the authors considered HER4 could homodimerize as well? It appears from the data presented in Figure 4 and S12 that the propensity to form homodimers is greater for HER4 than to heterodimerize with HER2, despite the VR/IQ substitutions. This also appears to be supported by the reasonable amount of signal for pERK in lanes with HER4-IQ alone in the presence of NRG1b. It is recommended that the authors comment on this possibility.

The IQ mutation, originally engineered to disrupt the receiver interface in EGFR, has been shown to have residual activity, which is greater than the mutation on the opposite site of the asymmetric dimer interface (VR) (PMID:16777603). This might be because this mutation partially destabilizes an inactive state of HER kinases by disrupting the hydrophobic interactions, which are both important for kinase inhibition and for stabilization of the active dimer. While IQ mutation is significantly inhibitory, as evidenced by the fact that we do not detect NRG1b-dependent HER4 phosphorylation in cells expressing HER4-IQ alone, it is possible that undetectable levels of phosphorylated HER4 cause the small increase in pERK signal. To acknowledge this possibility, we added the following sentence to the appropriate paragraph on page 10 in the main text:

“Small increases in pERK levels in cells expressing the HER4-IQ construct are consistent with previous observations that the IQ mutation in HER kinase domains has small residual activity through homodimerization (PMID:16777603).”

5. In the following line, "NRG1b-induced phosphorylation of HER2, HER4, ERK and AKT was not notably affected by substitution of the HER4 dimerization arm to a GS-arm relative to wild type receptors", it is unclear what the authors mean by wild-type receptors? There is presently no wildtype HER2 and/or HER4 tested in this blot.

We thank the reviewer for pointing this out. Wild type receptors here refer to WT dimerization arm sequences in contrast to GS-arm mutants. We corrected the language in the appropriate place in the main text:

“NRG1b-induced phosphorylation of HER2, HER4, ERK and AKT was not notably affected by substitution of the HER4 dimerization arm to a GS-arm relative to receptors featuring wild type dimerization arm sequences, indicating that the HER4 dimerization arm is not required for assembly and activation of HER2/HER4 heterodimers (Figure 2e).”

6. Considering the asparagine residues can potentially mediate stabilization of HER2-HER4 dimers through glycosylation, the authors should include western blot data for receptor-activation for mutants where glycosylation can be disrupted. This could minimally instruct the reader on how functionally relevant the identified interactions like N576-N358 are.

We agree with the Reviewer that this is a very interesting and important point, and it is subject of our future investigations. The different spectra of glycosylation that we observe between HER4 homodimers and HER2/HER4 heterodimers suggest that glycans will modulate these interactions differently. We speculate that glycans will likely be more important for HER4 homodimerization where glycosylation is more pronounced in our reconstructions. To investigate how these interactions change in the absence of single glycan modifications or their combinations, will also require taking into consideration how glycan mutations will alter an equilibrium between HER4 homodimers and HER2/HER4 heterodimerization. Such studies will require months of mutagenesis and optimization of controlled expression of such mutants, ideally generation of stable cell lines, and likely and ideally structural follow up studies. We respectfully argue that this undertaking is beyond the main scope of the current manuscript, and conceptually constitutes a separate, very important question that we are working on.

**Reviewer #1 (Recommendations For The Authors):**
The structural coordinates should be deposited in the RCSB.

The coordinates are deposited in the RCSB:

https://www.rcsb.org/structure/8U4L

https://www.rcsb.org/structure/8U4K

https://www.rcsb.org/structure/8U4I

https://www.rcsb.org/structure/8U4J

**Reviewer #2 (Recommendations For The Authors):**
1. Figure S1b authors should ideally include a silver stain gel to assess the purity of the heterodimer-ligand complex. Although HER subunits are discernible, there is no clear band for NRG1b.

Given its small size (9.7 kDa) our NRG1b construct is typically difficult to detect in our samples, but we would like to respectfully argue that the fact that we can resolve it at high resolution in our cryo-EM reconstructions provides sufficient evidence that it is present. Likewise, we argue that the Coomassie-stained gel we present in the manuscript is sufficient. It demonstrates that our purifications yield a stoichiometric complex of enough purity to obtain a high resolution cryo-EM reconstruction. Since we are not making any other claims about these preparations, we respectfully argue that providing a silver stain gel is not necessary to support conclusions of our study.

We thank the reviewer for point this out. To best reflect what we wanted to convey, we change it to: “and is the same as observed in structures of an isolated HER2 ectodomain.”

2. Page 8 first paragraph line 3, although one can deduce where the ligand binding pocket is, it would be clearer if this is marked in Figure 1d.

We added arrows in the figure to indicate the ligand-binding pocket.

3. Figure 2b inset A needs to be labeled 'A'.

The inset was already labelled but in a different corner. We rearranged the label to make it clearer.

4. Figure S5c will benefit from inset images zooming into the dimerization arm. It is hard to visualize the subtleties of the structural changes in the current format.

Figure 5c predominantly shows side-views of various heterodimer overlays to highlight subtle differences in larger-scale assembly that correlate with differences in dimerization arm engagement. This side-orientation is not suitable for zooming into the dimerization arm regions, which can only be effectively visualized in front views (the view of the heart-shaped dimer illustrated in Figure 1a). We show a zoomed-in view of this representation in main Figure 2c, which is what we understand the Reviewer is requesting.

5. Fig 3e is it A102 or A202 in the bottom-most panel.

This is now corrected, thank you.

6. Fig S9 revisit the color code for NRG1b, it appears there is no blue subunit of NRG1b. Also revisit the RMSD in the figure legend, since the text appears to suggest a different set of RMSDs for the 3 overlays.

We fixed the color code in the Figure, thank you.

In reference to Figure S9 (Figure S11 in the revised manuscript) we discuss two types of RMSDs:

1. RMSDs between our cryo-EM homodimers and the crystal structure homodimers. The structure overlays are shown in Figure S9a and RMSD values were mentioned in the Figure legends. However, in the original manuscript we did not explicitly mention these values in the main text but have now added them to the main text of the revised version of the manuscript.

2. RMSDs between monomers within our cryo-EM structures and within monomers of the crystal structure. Figure S11b and Figure S11c of the revised manuscript show these overlays for the cryo-EM structures only and the values are present in the Figure legend. We do not show the respective overlay for the crystal structures, which is why the values are not mentioned in the Figure legends, but we discuss the values in the main text.

We recognize that this is confusing and added RMSD values for 1. to the main text and discuss this more carefully:

“Our cryo-EM structures of the HER4/NRG1b homodimer differs slightly from the three HER4/NRG1b homodimers per asymmetric unit in the 3U7U crystal structure in which each monomer adopts a different orientation of the domain IV relative to the rest of the ectodomain (Figure S9a, RMSD: 5.438 Å, 5.435 Å and 3.662 Å). Notably, our two cryo-EM HER4 homodimer structures are more symmetric than the crystal structures of the HER4/NRG1β ectodomain homodimer. RMSDs for monomers within our cryo-EM structures are 1.42 Å in the cryo-EM HER4/NRG1b homodimer and 1.58 Å in the HER4/BTC homodimer (Figure S9b+c) compared to the monomers in the crystal structures which align with RMSDs of 1.67 Å, 5.76 Å and 2.38 Å”

7. Page 12 paragraph 2 last line, expand on the abbreviation NAG.

It is now expanded.

8. What is the slit width used for the energy filter during data collection?

The slit width was 20 eV. We added this information to the Methods section.

9. The crosslinking conditions of 0.2% glutaraldehyde for 40 min on ice, with no quenching seems rather harsh. Have the authors attempted other crosslinking conditions? Do milder conditions or GraFix not help with complex stabilization?

We thank the Reviewer for pointing this out. The reaction was quenched after 40 min by addition of 40 µl of 1M Tris pH 7.4 buffer. This information is now included in the Methods section. We have screened ideal crosslinking conditions for HER4 homodimers, and previously for HER2/HER3 heterodimers, and found that these crosslinking conditions were the mildest conditions that achieved complete crosslinking as assessed by SDS-PAGE.

10. Have the authors used default parameters for all their data processing steps? Were additional steps like local per-particle CTF refinement and global defocus refinement employed during refinement?

We did not perform any per particle CTF refinements as we previously have not observed any improvement from running such refinement on our size particles on top of per patch CTF estimation that already takes into account local CTF differences per micrograph. To make the manuscript clearer in this regard we added the following statement to the Methods section: “Unless specifically mentioned here or in the processing workflow, default parameters in CryoSPARC were used for each processing step.”